# Step-wise evolution of azole resistance through copy number variation followed by *KSR1* loss of heterozygosity in *Candida albicans*

Pétra Vande Zande[1], Cécile Gautier[2], Nora Kawar[3], Corinne Maufrais[2,4], Katura Metzner[1], Elizabeth Wash[1], Annette K. Beach[1], Ryan Bracken[1], Eli Isael Maciel[2], Nívea Pereira de Sá[5], Caroline Mota Fernandes[5], Norma V. Solis[6], Maurizio Del Poeta[5,7,8], Scott G. Filler[6,9], Judith Berman[3], Iuliana V. Ene[2], Anna Selmecki[1]*

**1** Department of Microbiology and Immunology, University of Minnesota, Minneapolis, Minnesota, United States of America, **2** Institut Pasteur, Université Paris Cité, Fungal Heterogeneity Group, Paris, France, **3** Shmunis School of Biotechnology and Cancer Research, George S. Wise Faculty of Life Sciences, Tel Aviv University, Ramat Aviv, Israel, **4** Institut Pasteur Bioinformatic Hub, Université Paris Cité, Paris, France, **5** Department of Microbiology and Immunology, Stony Brook University, Stony Brook, New York, United States of America, **6** Division of Infectious Diseases, Lundquist Institute for Biomedical Innovation at Harbor UCLA Medical Center, Torrance, California, United States of America, **7** Division of Infectious Diseases, School of Medicine, Stony Brook University, Stony Brook, New York, United States of America, **8** Veterans Administration Medical Center, Northport, New York, United States of America, **9** David Geffen School of Medicine at UCLA, Los Angeles, California, United States of America

* selmecki@umn.edu

**Data Availability Statement:** RNA sequences reported in this paper have been deposited in the NCBI Sequence Read Archive, https://www.ncbi.

## Abstract

Antimicrobial drug resistance poses a global health threat, requiring a deeper understanding of the evolutionary processes that lead to its emergence in pathogens. Complex evolutionary dynamics involve multiple mutations that can result in cooperative or competitive (clonal interference) effects. *Candida albicans*, a major fungal pathogen, displays high rates of copy number variation (CNV) and loss of heterozygosity (LOH). CNV and LOH events involve large numbers of genes and could synergize during evolutionary adaptation. Understanding the contributions of CNV and LOH to antifungal drug adaptation is challenging, especially in the context of whole-population genome sequencing. Here, we document the sequential evolution of fluconazole tolerance and then resistance in a *C. albicans* isolate involving an initial CNV on chromosome 4, followed by an LOH on chromosome R that involves *KSR1*. Similar LOH events involving *KSR1*, which encodes a reductase in the sphingolipid biosynthesis pathway, were also detected in independently evolved fluconazole resistant isolates. We dissect the specific *KSR1* codons that affect fluconazole resistance and tolerance. The combination of the chromosome 4 CNV and *KSR1* LOH results in a >500-fold decrease in azole susceptibility relative to the progenitor, illustrating a compelling example of rapid, yet step-wise, interplay between CNV and LOH in drug resistance evolution.

nlm.nih.gov/bioproject (BioProject ID PRJNA1063495), and whole genome sequencing data for strains described in this paper (S1 Table) have been deposited in NCBI Sequence Read Archive (BioProject ID PRJNA1071177).

**Funding:** This research was supported by the National Institute of Allergy and Infectious Diseases grant R01AI143689 and Burroughs Wellcome Fund Investigator in the Pathogenesis of Infectious Diseases Award No 1020388 to A.S., the European Research Council under the European Union's Horizon 2020 research and innovation programme grant agreement No 951475 to J.B., the ANR GENOMEHET grant and a PTR Carnot Pasteur grant to I.V.E., the National Institute of Allergy and Infectious Diseases grant AI125770 and the Research Career Scientist Award No IK6 BX005386 from the U.S. Department of Veterans Affairs to M. D.P. P.V.Z. is a Fellow of The Jane Coffin Childs Memorial Fund for Medical Research, N.K. was supported by an Israel Higher Education Committee Fellowship for Arab students, E.I.M was supported by an FRM Espoirs de la Recherche Postdoctoral Fellowship, and I.V.E. is a CIFAR Azrieli Global Scholar in the CIFAR Program Fungal Kingdom: Threats & Opportunities. The Minnesota Supercomputing Institute at the University of Minnesota provided resources that contributed to the research results reported within this paper. The funders had no role in study design, data collection and analysis, decision to publish, or preparation of the manuscript.

**Competing interests:** Dr. MDP, M.D., is a Co-Founder and Chief Scientific Officer 1102 (CSO) of MicroRid Technologies Inc. The goal of MicroRid Technologies Inc. is to develop new anti-fungal agents of therapeutic use. All other authors declare no competing interests.

## Author summary

The rise of drug-resistant microbes is a global health crisis. To combat this, we need to understand how these organisms evolve drug resistance. This study focuses on *Candida albicans*, a common fungal pathogen, and its ability to develop resistance to fluconazole, a widely used antifungal drug. We observe *C. albicans* rapidly evolve drug resistance through a two-step process involving large-scale genetic changes. First, it undergoes a copy number amplification, then it undergoes a loss of heterozygosity at the *KSR1* gene, which is involved in sphingolipid biosynthesis. These changes work together to dramatically increase the fungus's ability to survive in the presence of the drug. This research highlights how quickly and efficiently yeast can adapt to overcome antifungal drug and emphasizes the importance of studying large-scale genetic changes, not just individual mutations, when investigating drug resistance. Our findings could help in developing new strategies to predict and prevent the emergence of resistant pathogens, ultimately improving patient care in the face of this growing threat.

## Introduction

The evolutionary processes that lead to the emergence of antimicrobial drug resistance in pathogens pose a global health threat. Understanding the evolution of resistance is complicated by the fact that multiple mutations often arise in the same population and either compete with each other (clonal interference) or co-occur in the same genetic background, thereby contributing to resistance either additively or epistatically [1–4]. *Candida albicans*, a frequent commensal and pathogen of humans, undergoes high rates of copy number variation (CNV) and loss of heterozygosity (LOH) [5–8], which often affect many genes and can contribute to evolutionary adaptation. However, CNV and LOH can be difficult to identify in populations of cells via whole genome sequencing. Therefore, the dynamics by which CNV and LOH drive adaptation to drug stress, how the two types of mutations interact, and the mechanisms by which they influence drug resistance are not fully understood.

Ergosterol, the fungal analog of cholesterol, is a key regulator of membrane fluidity. Two of the three major classes of antifungal drugs target ergosterol directly or indirectly. Azole drugs target lanosterol 14-alpha demethylase, a cytochrome P450 that catalyzes a key step in the ergosterol biosynthesis pathway, which is encoded by the gene *ERG11* in *C. albicans*. Mutations that occur in *ERG11* itself [4,9,10], those which affect its expression level [11–14], or mutations in other genes in the ergosterol biosynthesis pathway such as *ERG3* [15–17] can confer azole drug resistance to *C. albicans* isolates and other *Candida* species. However, the most prevalent resistance mechanisms affect the expression or activity of drug efflux pumps [1,3,18–21], which affect intracellular drug levels and influence membrane homeostasis in ways that affect drug susceptibility [22,23]. Together with sphingolipids, ergosterol also contributes to lipid raft formation and membrane polarization. The polarization of the lipid membrane is necessary for filamentation [24], a key virulence trait, and lipid rafts are important for the proper localization and activation of membrane-associated proteins including efflux pumps [25]. Thus, azole resistance and tolerance are also influenced by sphingolipid biosynthesis [21,26].

Azole tolerance, in contrast to resistance, is defined as slower persistent growth that is independent of drug concentration. Changes in drug susceptibility that result in clinical resistance, or the ability to grow above a clinical breakpoint, are measured as changes in $MIC_{50}$, the drug

concentration that reduces growth by 50% or more at 24 h (Minimum Inhibitory Concentration, $MIC_{50}$) [27]. Tolerance is measured as the relative growth in supra-MIC concentrations of drug at 48 h (supra-MIC growth, SMG) [21,28]. Tolerance relies upon the robust function of many stress response pathways [28], and high tolerance has been associated with clinical persistence during treatment [28–31]. We do not yet understand how mutations conferring high tolerance impact treatment, and whether they might precede or be connected to the acquisition of drug resistance.

Multiple mechanisms could link sphingolipid biosynthesis (S1 Fig) to azole susceptibility and/or tolerance [26]. In general, mutations or chemical inhibitors that disrupt steps in the sphingolipid biosynthesis pathway cause increased susceptibility or act synergistically with azoles [32–38]. Despite its importance in the sphingolipid pathway [39], the 3-ketosphinganine reductase encoded by *KSR1* has not been previously connected to azole susceptibility. However, a *ksr1Δ/ksr1Δ* null mutant generated in a lab strain increased the fluconazole $MIC_{50}$ by 2-fold [39], providing initial evidence that changes to *KSR1* function could influence azole responses.

The heterozygosity of the *C. albicans* diploid genome serves as a rich source of recessive variants that can be revealed rapidly via LOH [8,40,41]. The *C. albicans* reference genome from isolate SC5314 contains almost 200 heterozygous nonsense alleles that encode early stop codons [41]; such alleles might limit the selection of LOH events if they are recessive lethal [42]. Yet heterozygous recessive alleles can also be advantageous when homozygosed via LOH events [43–45]. For example, homozygosis of a hyperactive allele of the efflux pump regulator Tac1p can confer high levels of azole resistance [19]. LOH in *C. albicans* frequently affects long portions of chromosome arms, encompassing thousands of heterozygous positions [46,47], thereby making it difficult to pinpoint the specific genes and nucleotides by which such events affect antifungal drug responses. As a result, despite their prevalence, the mechanisms by which most LOH events affect phenotypes remain unknown.

CNV is another important class of mutations frequently found in *C. albicans* strains that evolve under stress. Both whole-chromosome and segmental aneuploidy are associated with decreased susceptibility and/or increased tolerance to antifungal drugs in clinical isolates and in laboratory evolution experiments [48–50]. In some cases, the mechanisms causing decreased susceptibility via CNV have been identified. For example, amplification of the left arm of chromosome (Chr) 5 occurs via an isochromosome structure (i(5L)) that provides two additional copies of the fluconazole (FLC) target *ERG11* and of a transcriptional activator of drug efflux pumps, *TAC1*; the extra copies of these two genes largely explains the drug resistance in strains carrying the i(5L) karyotype [12,51]. While the specific mechanisms of resistance have not been elucidated for most CNVs, the recurrent association of decreased susceptibility or increased tolerance with specific CNVs suggests that specific genes within these amplified regions are responsible [48,49,52,53]. As with LOH, identifying the mechanisms of drug resistance is complicated by the large number (often hundreds) of genes amplified within a given CNV, any of which could potentially contribute to resistance via different mechanisms.

Here, we describe the step-wise evolution of an azole resistant isolate of *C. albicans* via an initial CNV on Chr4, followed by a small LOH on ChrR. These two events contribute additively to drug resistance and result in a 500-fold increase in final $MIC_{50}$ relative to the progenitor. Furthermore, we identify *NCP1*, a gene within the Chr4 CNV, whose overexpression results in a 2-fold increase in $MIC_{50}$. We also localize the specific nucleotides affected by the LOH on ChrR to the *KSR1* coding sequence and show that they are important for decreased susceptibility and increased tolerance. Combining the LOH in *KSR1* with the Chr4 CNV decreases susceptibility by >500-fold, providing a powerful example of rapid step-wise

evolution of *bona fide* drug resistance via multiple mechanisms. Furthermore, in the reference isolate SC5314, *KSR1* contains a heterozygous nucleotide that encodes an early stop codon; this nonsense allele results in a truncated protein that lacks a membrane localization domain. Remarkably, we identified four independent LOH events that each affect residues in *KSR1* but do not generate homozygous early stop codons. This highlights the power of LOH to recombine heterozygous variants and to de-couple linked variants in *C. albicans*.

## Results

### Step-wise acquisition of resistance during adaptation to fluconazole

We recently identified a lineage (AMS4058) in an evolution experiment in which the drug sensitive progenitor strain SC5314 acquired FLC resistance after 10 passages in 1 μg/mL FLC, resulting in an increase in $MIC_{50}$ from 0.5 μg/mL to >256 μg/mL FLC [49]. The AMS4058 lineage also had reduced filamentation *in vitro* and decreased virulence in a mouse model of disseminated candidiasis (S2A and S2B Fig). To understand the dynamics of adaptation in lineage AMS4058, we measured the $MIC_{50}$ at each of the 10 passages. The $MIC_{50}$ increased 2-fold after passage 1 and again after passage 3; and increased dramatically to >256 μg/mL FLC after passage 4 (Fig 1A). We also measured FLC tolerance after each passage (supra-MIC growth, SMG, see Methods) [21,28]. SMG increased dramatically after the first passage and steadily decreased in passages 2 and 3 (Fig 1B). We hypothesized that variants conferring large increases in tolerance were dominant early in the evolution experiment (passages 1 to 3), whereas variants conferring resistance arose around passage 4 and continued to increase in frequency in all subsequent passages.

### Aneuploidy and copy number variation arise early and yield decreased susceptibility and increased tolerance

To identify the genetic changes associated with decreased drug susceptibility and increased tolerance, we performed population-level whole genome sequencing of each passage of the evolution experiment. We first estimated the population-average copy number of each chromosome by examining sequencing read depth across the genome for all 10 passages. Increases in read depth indicative of copy number variants (CNVs) were present in the population from passage 2 to passage 10 (Fig 1C–1E). The median copy number of Chr3 was ~2.18, 2.27, and 2.15 for passages 1, 2, and 3, respectively (Fig 1D), suggesting that ~15–30% of the population was trisomic for Chr3 at these early passages with high tolerance (see Methods). Chr5 also rose to a frequency of ~2.17 by passage 2 (Fig 1D). Chr3 contains several genes encoding efflux pumps and their regulators, such as Cdr1, Cdr2, and Mrr1; and Chr5 contains genes encoding Erg11 (the azole drug target) and efflux pump regulator Tac1. Additional copies of these genes have the potential to contribute to decreased susceptibility and increased tolerance observed in the early populations of the evolution experiment.

In addition to evidence of aneuploidy, we detected a segmental amplification of Chr4 (Fig 1C and 1E), consisting of a stair-step structure with two flanking regions (~20–25 kb, A21 Chr4 coordinates ~532000 to ~553000, and ~681000 to ~704500) surrounding a central region with the highest copy number (~100 kb, A21 Chr4 coordinates ~553000 to ~674000) (Fig 1F). Each amplified region was flanked by inverted repeat sequences, consistent with segmental amplifications as previously described (Fig 1F, [52]). Median read depth in the central region increased from a population average relative copy number of 2 in passage 1 to ~3.5 by passage 2, ~5 by passage 3, and then gradually rose to ~6 copies by passage 10 (Fig 1E).

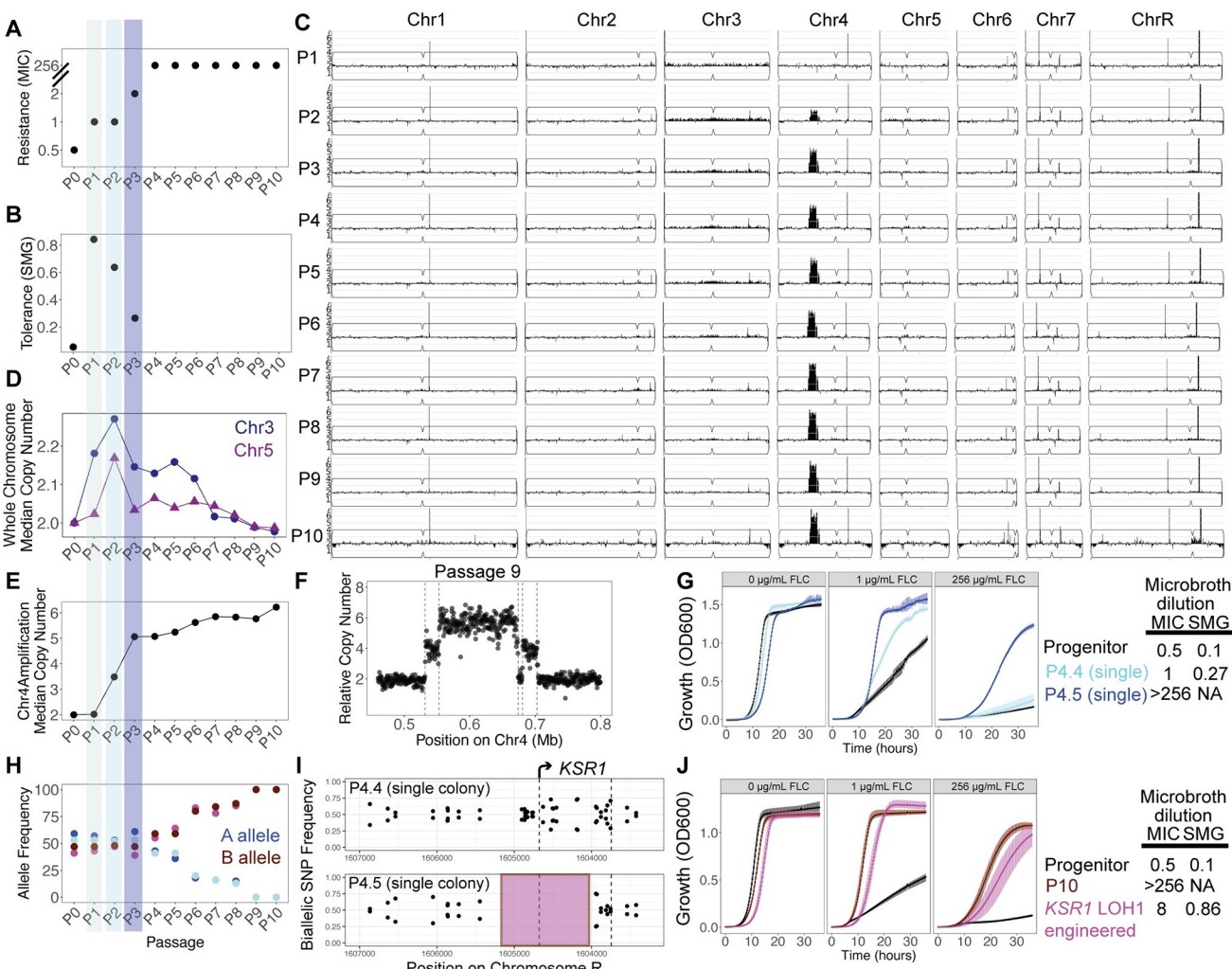

**Fig 1. Step-wise evolution of drug resistance via CNV and LOH.** (A) MIC$_{50}$, defined as the drug concentration at which growth is inhibited by >50%, is plotted on the y-axis for each passage of the evolution experiment. (B) SMG, defined as the average growth in drug concentrations greater than the MIC$_{50}$, is plotted on the y-axis for each passage of the evolution experiment. (C) Read depth from whole genome sequencing of the entire population for each passage of the evolution experiment is plotted according to genomic position and normalized to the average read depth for the entire genome. (D) Median copy number, calculated from read depth relative to the entire genome, for whole chromosomes with variation in copy number (Chr3 and Chr5) is plotted on the y-axis for the entire population at each passage of the evolution experiment. (E) Median copy number of the central amplified portion of Chr4, calculated from read depth in the entire passaged population relative to the rest of the genome, is plotted on the y-axis at each passage of the evolution experiment and connected by line segments. (F) Relative copy number for the entire population at passage 9 at the region containing a CNV, calculated from the relative average read depth, is plotted on the y-axis according to the position on Chr4 on the x-axis. Each point represents the average read depth of a 500 bp window. Dashed lines show the location of repeat sequences in the *C. albicans* genome [52]. (G) OD$_{600}$ values for liquid culture growth assays are plotted over time for the wild-type progenitor (black), a single colony containing the Chr4 amplification without LOH at *KSR1* (P4.4 single colony, light blue), and a single colony containing the Chr4 amplification with the LOH at *KSR1* (P4.5 single colony, dark blue). Growth in rich media, 1 μg/mL FLC, and 256 μg/mL FLC are shown. Error bars are standard errors for three replicates. Points are averages. A table to the right shows average MIC$_{50}$ and SMG values for the listed strains. (H) Allele frequencies for 2 single nucleotide variants at *KSR1* amino acid positions 189 and 192 are plotted for each passage of the evolution experiment. Reference "A" alleles are shown in cyan, and reference "B" alleles are shown in magenta. (I) Biallelic single nucleotide polymorphism (SNP) frequencies are plotted across the *KSR1* locus on chromosome R for two single colonies isolated from passage 4, with MIC$_{50}$ = 1 (P4.4, top) and MIC$_{50}$ >256 μg/mL FLC (P4.5, bottom). The absence of biallelic SNPs in P4.5 indicates a loss of heterozygosity highlighted by the pink-shaded region. (J) OD$_{600}$ values for liquid culture growth assays are plotted over time for the wild-type progenitor (black), the P10 evolved population (red), and a strain engineered to contain the LOH at *KSR1* (pink). Growth in rich media with 0 μg/mL, 1 μg/mL, or 256 μg/mL FLC are shown. Error bars are standard errors for three replicates. Points are averages. A table to the right shows MIC$_{50}$ and SMG values for the listed strains.

To identify individual genotypes within the passaged populations, we isolated and sequenced 5 single colonies each from passages 2, 3, and 4. Passage 2 included one colony with wild type chromosome copy numbers (euploid), one colony with an LOH of ChrR and duplication of the B homolog, resulting in a ChrR BB genotype, two colonies with monosomy of the right end of ChrR, and one colony with homozygosis of the left arm of Chr4 resulting in an AA genotype (S3A Fig). For each unique genotype from passage 2 we performed growth curve and microbroth dilution assays and found that the colony with ChrR LOH grew faster than the progenitor strain in all concentrations of FLC, with an $MIC_{50}$ of 2 μg/mL FLC and an SMG of 0.78 (S3B Fig). In passage 3 we found two distinct genotypes: 2 colonies displayed concurrent trisomy of Chr3 and Chr6 along with LOH encompassing the entirety of ChrR resulting in a ChrR AA genotype; and 3 colonies displayed the Chr4 segmental amplification described above (S3C Fig). Both genotypes from passage 3 also grew better than the progenitor in FLC in growth curve assays and had a slightly elevated $MIC_{50}$ of 1 μg/mL FLC and SMG of 0.25 and 0.3 (S3D Fig). Thus, early in the evolution experiment, different copy number variants arose and conferred a 2- to 4-fold increase in $MIC_{50}$ and increased tolerance.

The amplified region of Chr4 does not contain genes known to influence drug susceptibility upon overexpression. This region includes ~88 genes, including *NCP1*, which encodes the Erg11p reductase (S4A Fig). We tested the hypothesis that overexpression of Ncp1p is sufficient to promote growth in 1 μg/mL FLC by engineering strains with *NCP1* under the control of a tet-off promoter system inserted at one of the two copies of the native locus. Without the addition of doxycycline, these strains overexpressed *NCP1* at ~8-fold the level of the wild-type strain; with the addition of doxycycline, *NCP1* was expressed at 0.5-fold relative to its expression in wild-type cells (S4B Fig). This level of *NCP1* overexpression closely matched that of strains containing the Chr4 CNV in FLC, which was also ~8-fold relative to the wild-type strain (S4C Fig). Overexpression of *NCP1* in these engineered strains resulted in a 2-fold increase in $MIC_{50}$ that was abolished with the addition of doxycycline (S4D Fig). Because there is allelic variation at the *NCP1* locus in SC5314, we overexpressed the A and B alleles independently; both *NCP1* alleles resulted in the same 2-fold increase in $MIC_{50}$ (S1 Table). Thus, overexpression of *NCP1* is sufficient to increase growth in low drug concentrations and provides at least one explanation for the increased $MIC_{50}$ in the single colonies carrying the Chr4 CNV. However, overexpression of *NCP1* cannot explain the $MIC_{50}$ of >256 μg/mL FLC displayed by passages 4–10 of the evolution experiment.

## LOH at *KSR1* produces a resistance phenotype

To identify the genetic changes responsible for the resistance seen in passages 4–10, we examined the genomes of 5 single colonies from passage 4 and measured their growth across a range of FLC concentrations. All 5 colonies contained the Chr4 CNV (S3E Fig); however, only 1 of the 5 colonies, isolate P4.5, grew significantly faster than the progenitor in 256 μg/mL FLC and had an $MIC_{50}$ of >256 μg/mL FLC (S1 Table and Fig 1G). Interestingly, P4.5 contained a short (711 bp) LOH of the ChrR B homolog between coordinates 1604191 to 1604902 that was not present in the other colonies. Population-level sequencing of each passage showed that allele ratios in this region were biased for allele A in passages 1–3 but switched to a bias for allele B in passage 4 and steadily rose to 100% frequency of the B allele by passage 9 (Fig 1H). This ~700 bp LOH (which we will now call LOH1) encompassed ~⅔ of *KSR1* (CR_07380C) which encodes 3-ketosphinganine reductase (Fig 1I) and catalyzes the second step in sphingolipid biosynthesis [39].

To directly test the effect of the 711 bp LOH at *KSR1* (*KSR1* LOH1) on drug susceptibility, we replaced this portion of the *KSR1A* allele with *KSR1B*, reconstructing the evolved LOH in

the progenitor background (SC5314) (without the Chr4 CNV). This strain, (*KSR1* LOH1) grew faster than the progenitor at FLC concentrations from 1 to 256 μg/mL, with an $MIC_{50}$ of 8 μg/mL FLC and an SMG of 0.86 (Fig 1J). Thus, *KSR1* LOH1 alone is sufficient to cause a 16-fold decrease in susceptibility and a high tolerance phenotype (~8-fold higher than the progenitor).

Next, we tested the hypothesis that overexpression of *NCP1* and the *KSR1* LOH1 in combination could explain the evolved resistance phenotype. A strain engineered to carry both the *tetO-NCP1* overexpression construct and the *KSR1* LOH1 genotype (tetO-*NCP1*, *KSR1* LOH1) grew faster than either single mutant alone in low and high FLC. However, the strain with tetO-*NCP1* and *KSR1* LOH1 had a final $MIC_{50}$ of 4 and SMG of 0.66 (S5 Fig), which is not sufficient to reproduce the $MIC_{50}$ of single colony P4.5, which carried the ~150kb Chr4 CNV and the *KSR1* LOH1. This implies that one or more additional genes within the Chr4 CNV might be necessary, in combination with the *KSR1* LOH1, to account for the high $MIC_{50}$ of the populations at passages 4–10 and of single colony P4.5. Additionally, amplification of the Chr4 CNV coincided with LOH of Chr4 sequences that flank the CNV (S2C Fig). Therefore, we cannot rule out the possibility that LOH of other genes on Chr4 could contribute to high resistance in strain P4.5 as well.

To ask if Ksr1p is necessary for drug resistance, we tried to engineer null strains lacking both copies of *KSR1*. We obtained heterozygous mutants by deleting either the A or B allele individually. Strains lacking the *KSR1* B allele had a wild-type $MIC_{50}$ of 0.5 μg/mL FLC, while strains lacking the *KSR1* A allele had an $MIC_{50}$ of 2 μg/ml FLC, a 4-fold increase relative to the parent strain (S6A Fig). Despite several attempts in multiple labs, the only two *ksr1Δ/Δ* transformants recovered had undergone large LOHs encompassing most of ChrR (S6B Fig), which confounded the interpretation of phenotypes associated with the complete loss of Ksr1p. In *Saccharomyces cerevisiae*, the *KSR1* ortholog (*TSC10*) is essential [54]; however, in *C. albicans*, null mutants of *KSR1* (strains SC90, SC91) were previously described in strain BWP17, an auxotrophic derivative of SC5314 [39]. Our whole genome sequencing of these original BWP17 *ksr1Δ/Δ* null strains revealed that both mutants had also undergone LOH encompassing most of ChrR (S6B Fig). Interestingly, two of the four *ksr1Δ/Δ* null strains were homozygous for ChrR AA (blue) and the other two were homozygous for ChrR BB (pink), but in all four strains the region flanking the *KSR1* locus was homozygous for the A allele (AA) (S6C Fig). Importantly, other manipulations of *KSR1* usually were not accompanied by large LOHs (see below), suggesting that *KSR1* might be essential in the SC5314 background when ChrR maintains normal heterozygosity.

## *KSR1* LOH1 results in reduced sphingolipids, lower intracellular FLC, higher drug efflux pump activity, and decreased toxic sterol content

Ksr1p catalyzes an early step in sphingolipid biosynthesis (S1 Fig) and changes in its function may alter the cellular sphingolipid composition [39,55,56]. Consistent with this, *ksr1Δ/Δ* strains (SC90 and SC91, containing large ChrR LOH regions) had lower levels of inositol phosphorylceramides and were sensitized to inhibitors of subsequent steps of sphingolipid biosynthesis [39]. To determine the effect of the evolved *KSR1* LOH1 we compared the lipid content of the progenitor (SC5314) and *KSR1* LOH1 (engineered into SC5314), with and without FLC treatment, using mass spectrometry. In the absence of FLC, the abundance of all sphingolipid species was reduced in the *KSR1* LOH1 strain relative to the progenitor (Fig 2A). In the progenitor, upon FLC exposure (1 and 10 μg/mL) intermediates of the sphingolipid pathway increased and glucosylceramides decreased in abundance (Fig 2A). By contrast, the *KSR1* LOH1 strain had consistently low levels of both phytosphingolipids and glucosylceramides. In

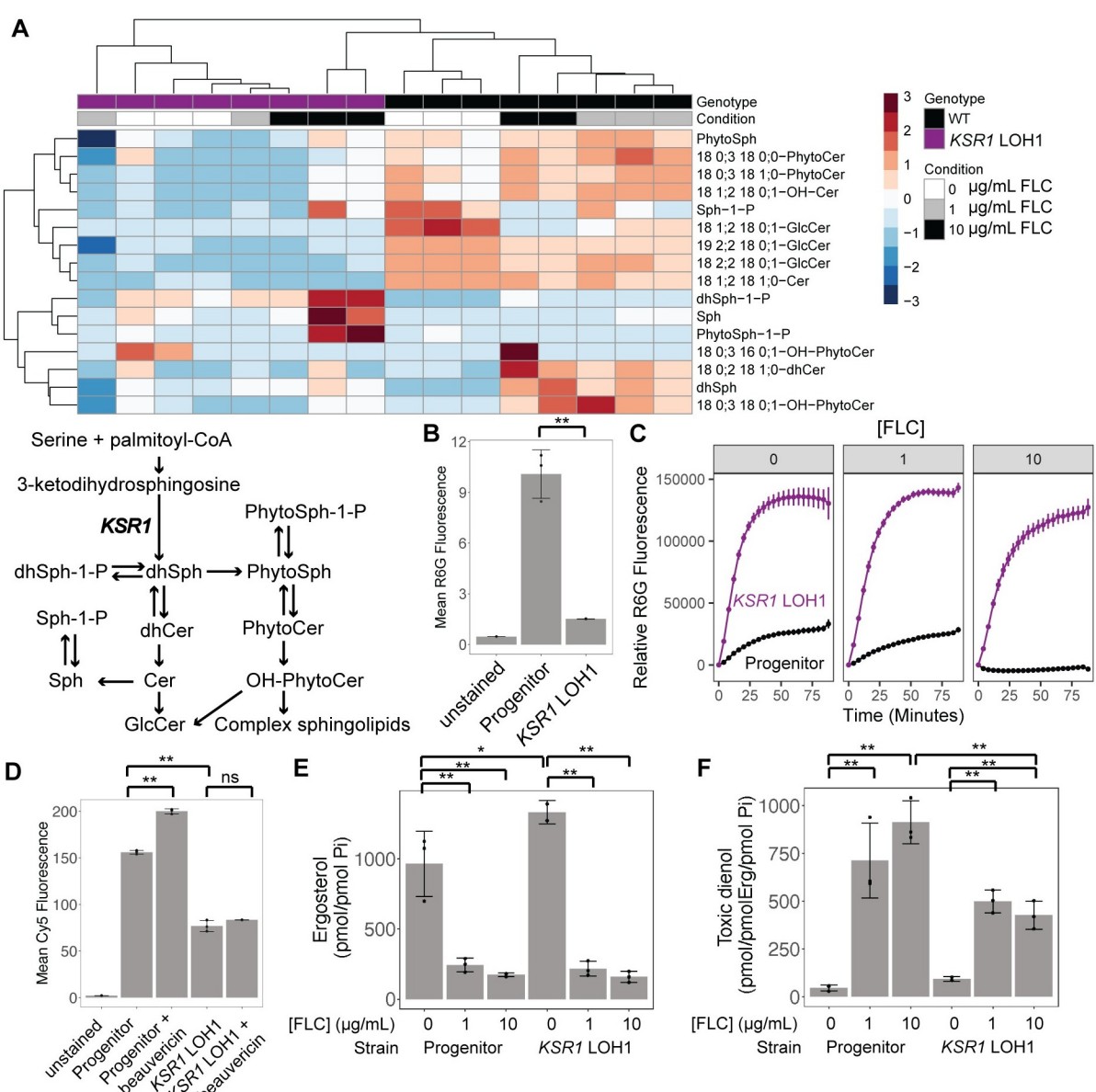

**Fig 2. LOH at *KSR1* affects cellular lipid profiles, intracellular FLC concentrations, drug efflux, and sterol levels.** (A) A heatmap shows abundance of sphingolipid species relative to inorganic phosphate present in each strain and condition relative to the mean of all samples for each respective lipid species. Heatmap colors indicate row-wise z-scores, red being high and blue being low. Rows are lipid species and columns are individual samples, both of which are clustered by k-means clustering as shown by the dendrograms on the top and left. Columns include 3 replicates each of the wild type progenitor strain (black) grown in 0 μg/mL (white), 1 μg/mL (grey), and 10 μg/mL FLC (black), and the *KSR1* LOH1 engineered strain (purple) in the same conditions. Two outlier samples were removed (WT in 10 μg/mL FLC and *KSR1* LOH1 in 1 μg/mL FLC, see Methods). Wild type samples cluster according to condition, indicating consistent changes in sphingolipid species as FLC concentrations are increased. *KSR1* LOH1 samples do not consistently cluster by condition, indicating only subtle changes in sphingolipid species as FLC concentration increases. An abbreviated sphingolipid biosynthesis pathway is included for reference (see full pathway at S1 Fig). (B) Mean intracellular Rhodamine 6G (R6G) fluorescence, as measured by flow cytometry, is plotted with both points and bars for each strain listed after 4 hours of incubation with R6G. Each point represents one replicate, bars are means and whiskers are quartiles. (**Indicates a paired t-test p-value of $< 0.01$) (C) Extracellular R6G fluorescence is plotted on the y-axis over time (on the x-axis) after addition of glucose, for the progenitor strain (black) and engineered *KSR1* LOH1 (purple). Points are means and error bars are standard errors of the mean across 3 replicates. (D) As in (B), mean intracellular Cy5 (FLC-Cy5) fluorescence is plotted for each strain listed after incubation with FLC-Cy5 for 4 hours, with or without the addition of beauvericin. (E) Ergosterol content relative to inorganic phosphate as measured by GC-MS are shown for the progenitor strain and *KSR1* LOH1 strain after exposure to 0, 1, and 10 μg/mL FLC for 4 hours. (F) Abundance of the toxic dienol (14α-methylergosta-8,24(28)-dien-3β,6α-diol, peak at 24.3 min retention time) relative to ergosterol abundance is shown for the same samples shown in (E). (D and E) **Indicates a paired t-test p-value of $< 0.05$, *Indicates a p-value of $< 0.1$. Each point represents one replicate, bars are means and whiskers are quartiles for 3 biological replicates.

addition, exposure of the *KSR1* LOH1 strain to 10 μg/mL FLC resulted in increased sphingosine (Sph), phosphorylated dihydrosphingosine (dhSph-1-p) and phosphorylated phytosphingosine (PhytoSph-1-p) relative to no drug conditions, in two of the three replicates (Fig 2A). No such change in sphingolipid species was seen in the progenitor. Together, these results indicate that in general, sphingolipid species are decreased in the *KSR1* LOH1 strain relative to the progenitor strain, and that the two strains respond differently to FLC.

A reduction in sphingolipid species could significantly alter membrane structure, membrane fluidity, and the formation of lipid rafts, which help properly position membrane-associated proteins such as the drug efflux pump Cdr1p [25]. In addition, PhytoSph-1-P is thought to be a signaling molecule that can lead to the upregulation of drug efflux pumps such as Cdr1p and Cdr2p and a reduction in intracellular FLC [34]. Thus, changes in the sphingolipid composition and structure of the cell membrane might influence the activity of drug efflux pumps.

To evaluate the effect of *KSR1* LOH1 on efflux pump activity, we first monitored intracellular levels of Rhodamine 6G (R6G), a fluorescent molecule that is actively effluxed by ABC transporters [57], by flow cytometry after incubation for 4 hours. *KSR1* LOH1 intracellular R6G levels were ~3-fold lower than in the progenitor strain (Fig 2B, paired t-test p-value = 0.009). To measure the dynamics of active drug efflux, we performed a time-course in which cells were preloaded with R6G, and fluorescence of the supernatant was measured over time after the addition of glucose. The *KSR1* LOH1 strain had a ~6-fold increase in the rate of efflux relative to the progenitor strain after growth in 0, 1, and 10 μg/mL FLC (Fig 2C). In addition, using RT-qPCR, we found that, relative to the progenitor, *CDR1* was upregulated ~1.5 fold in the absence of FLC, and ~2 fold in 1 μg/mL FLC (S7 Fig). While this increase in expression likely contributes to the higher rate of efflux, the much more dramatic increase in efflux (~6-fold increase in rate, Fig 2C) suggests that additional mechanisms influence efflux activity. We next tested whether this increased efflux affected intracellular drug concentrations of cells after incubation for 4 hours with a fluorescently labeled FLC probe (FLC-Cy5, [58]), in the presence and absence of the ABC efflux pump inhibitor beauvericin. The *KSR1* LOH1 strain had significantly lower intracellular levels of FLC-Cy5 than the progenitor strain at 4 hours (Fig 2D, paired t-test p-value = 0.004). However, intracellular levels of FLC-Cy5 remained significantly lower than the progenitor strain in the presence of the efflux pump inhibitor beauvericin (Fig 2D), suggesting that efflux by ABC transporters alone cannot account for the differences in intracellular FLC levels. Together, these results indicate that cells with the *KSR1* LOH1 have higher rates of drug efflux and accumulate lower levels of intracellular FLC than the progenitor strain.

Increased activity of drug efflux pumps, likely resulting from the changes to the sphingolipid components of the membrane, can elevate cellular ergosterol levels even in the absence of FLC [22,23]. Therefore, we assayed sterol levels in progenitor and *KSR1* LOH1 cells in the presence of 0, 1, and 10 μg/mL FLC to determine the effect of the LOH on ergosterol and other sterols. We found that in the absence of drug, the *KSR1* LOH1 strain had a significantly higher ergosterol abundance than the progenitor (Fig 2E, two-sided t-test p-value = 0.099). The *KSR1* LOH1 strain also showed increased sensitivity to Amphotericin B, which targets membrane ergosterol (S8 Fig). In the presence of both 1 μg/mL and 10 μg/mL FLC, however, ergosterol abundance was significantly decreased to similar levels in both the progenitor and *KSR1* LOH1 strains (Fig 2E, p-values left to right: 0.028, 0.027, 0.009, 0.015). This suggests that reduced intracellular levels of FLC are not associated with increased ergosterol in the *KSR1* LOH1 strain. Rather, elevated efflux pump activity is indicative of changes to membrane homeostasis that make cells less sensitive to FLC. In agreement with this, the abundance of a toxic dienol (14α-methylergosta-8,24(28)-dien-3β,6α-diol) known to inhibit growth [59]

relative to ergosterol levels was significantly lower in *KSR1* LOH1 cells compared to progenitor cells in the presence of 10 μg/mL FLC (Fig 2F, two-sided t-test p-value = 0.0053). Together these results indicate that the *KSR1* LOH1 results in a reduction in sphingolipid species and changes to membrane homeostasis, which are accompanied by increased drug efflux pump activity and lower levels of the toxic dienol in the presence of FLC.

## Homozygosis of arginine at Ksr1p aa189 is the major driver of decreased drug susceptibility

To examine the effects of specific amino acid(s) in Ksr1p that contribute to the drug susceptibility phenotype, we compared *KSR1* LOH1 to three other independently evolved SC5314-strains that acquired *KSR1* LOH events during adaptation to 1 μg/mL FLC (Evolved LOH2, Evolved LOH3, Evolved LOH4, see Methods). These three evolved LOH events ranged in size from ~20 bp to ~2,000 bp of ChrR and each included a portion of the *KSR1* locus (Figs 3A and S9A and S1 Table). All three evolved strains had 4-fold increases in MIC$_{50}$ and

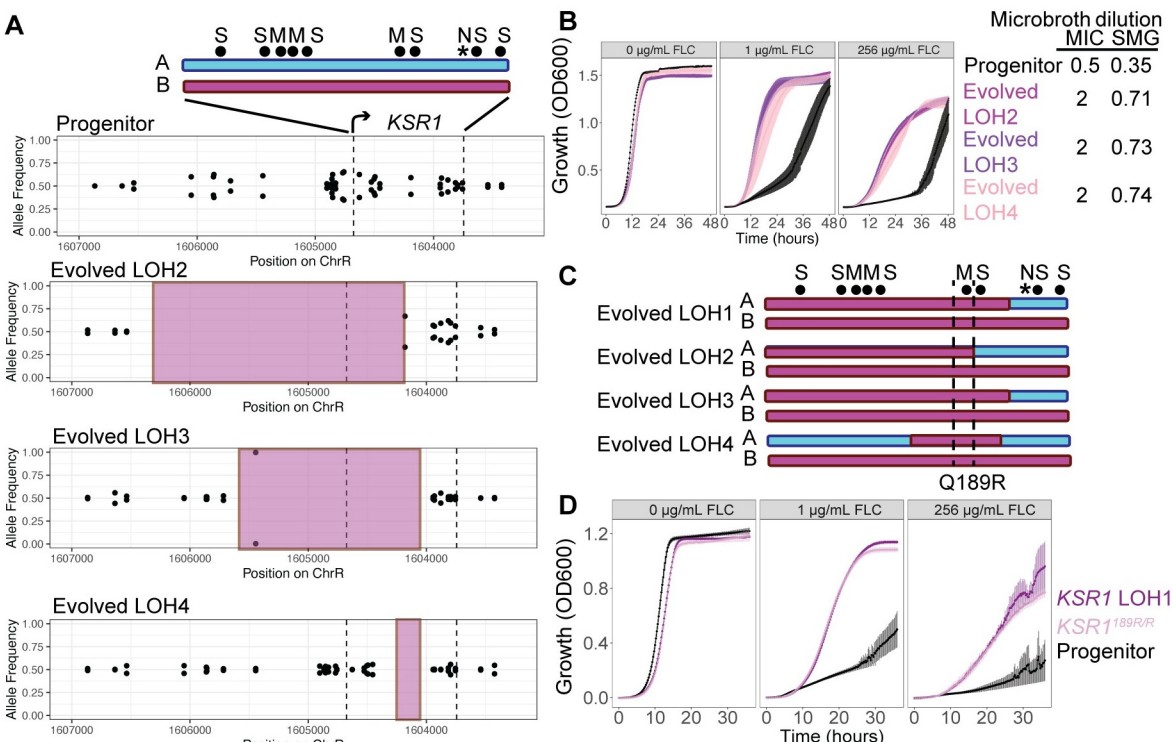

**Fig 3. LOH resulting in 189R/R produces a strong decrease in drug susceptibility.** (A) A schematic of the *KSR1* locus including heterozygous positions (black dots) that are either synonymous (S), missense (M), or nonsense (N, *) in the B allele relative to the A allele. A allele is shown in blue and B allele is shown in pink. Allele frequencies are plotted across the *KSR1* locus and a larger flanking region on chromosome R for the wild type progenitor (top) and the three evolved isolates (Evolved LOH2, Evolved LOH3, and Evolved LOH4). The absence of biallelic SNVs in the evolved isolates, or allele frequencies of 1, indicates a loss of heterozygosity in that region highlighted by the pink shaded region. Note that the chromosome coordinates are inverted on the map because the *KSR1* coding sequence is on the '*Crick*' strand of the chromosome. (B) OD$_{600}$ values for liquid culture growth assays are plotted over time for the wild type progenitor (SC5314, black), and the three evolved strains with LOH including some portion of the *KSR1* locus (Evolved LOH2, Evolved LOH3, and Evolved LOH4). Growth in rich media with 0 μg/mL, 1 μg/mL, and 256 μg/mL FLC are shown. Error bars are standard errors for three replicates. MIC$_{50}$ and SMG values are shown in a table to the right. (C) Schematics show portions of the *KSR1* coding region that underwent LOH in the four evolved LOH events. Dotted lines demarcate the single nonsynonymous SNV that was homozygous in all four LOH isolates, a nucleotide coding for glutamine in allele A and arginine in allele B at amino acid position 189. (D) Liquid growth assays, plotted as in (B), are shown for the progenitor strain (black), the strain engineered to contain the *KSR1* LOH1 (purple), and the *KSR1*$^{189R/R}$ engineered strain (pink).

increased SMG relative to the progenitor strain (Fig 3B). They also were defective in filamentation and had decreased virulence in a *Galleria mellonella* model of systemic fungal infection (S9 Fig).

RNA-sequencing of these three evolved strains when exposed to 0, 2, or 64 μg/mL FLC concentrations was analyzed for principal components. The first principal component, explaining the majority of the variance in the data (83%), separated samples by FLC treatment (S10A Fig). The second principal component, explaining 5% of the variance in the data, separated strains that were exposed to FLC by genotype (S10B Fig). This indicates that both wild type and evolved strains undergo major transcriptional changes upon exposure to FLC, and that, in the absence of FLC, transcriptional differences between progenitor and evolved strains are minimal. Expression levels of genes annotated as related to sphingolipid biosynthesis showed a similar trend, with samples clustering according to FLC exposure first, and then clustering weakly by genotype (S10C Fig). Although 5 genes catalyzing steps in sphingolipid biosynthesis were differentially expressed in all three evolved strains relative to the progenitor strain, no strong correlation between transcriptional changes and changes in corresponding sphingolipid metabolites was evident (S10D Fig), likely because most upregulated genes were downstream of *KSR1* in the biosynthesis pathway (S1 Fig). Because the *KSR1* LOH1 strain had high levels of glucose-dependent efflux in the R6G assay (Fig 2C), we also examined the expression of ABC drug efflux pumps. Efflux pumps *CDR1* and *CDR2* were not significantly upregulated in the three evolved strains (S10E Fig). *SNQ2*, which is annotated as a multidrug transporter in *S. cerevisiae* [60], was upregulated in all three strains (S10E Fig). Taken together, these data indicate that LOH affecting Ksr1p has a small effect on the global transcriptome and likely impacts the activity and/or localization of drug efflux pumps post-transcriptionally, and that the progenitor and evolved strains respond differently to FLC exposure.

In SC5314, the *KSR1* protein coding region harbors 6 synonymous and 4 nonsynonymous heterozygous nucleotides. The LOH events in all four FLC-evolved LOH strains became homozygous for the B allele at a single nonsynonymous variant encoding aa189 (Fig 3C). In the wild-type SC5314, allele A encodes glutamine and allele B encodes arginine ($KSR1^{189Q/R}$); in all four evolved LOH isolates (LOH1-LOH4) the sequence encodes arginine at aa189 ($KSR1^{189R/R}$). To test the effect of aa189R/R alone, we constructed a strain encoding $KSR1^{189R/R}$ and measured its growth in a range of FLC concentrations. The $KSR1^{189R/R}$ strain grew better than the progenitor across all FLC concentrations, with growth dynamics similar to those of the *KSR1* LOH1 strain and had the same $MIC_{50}$ and SMG values (Fig 3D). Thus, $KSR1^{189R/R}$ is a major driver of the decreased drug susceptibility in all four evolved strains (Fig 3D).

## Recurrent and precise LOH breakpoint unlinks the effect of 189R/R from the masking effect of the nonsense codon

*KSR1* codon 272 in the SC5314 progenitor encodes arginine on the A allele and encodes a nonsense mutation on the B allele ($KSR1^{272R/*}$). Notably, all four LOH events in FLC-resistant strains remained heterozygous at this codon ($KSR1^{272R/*}$, Fig 3C). The precision of the right LOH breakpoint between aa189 and aa272 in all four evolved strains could indicate strong selective pressure against homozygosis of the nonsense codon at aa272. We asked if a strain carrying the homozygous nonsense allele alone would be viable, and if so, if the nonsense allele would mask the effect of aa189R/R on drug susceptibility. To address these questions, we engineered mutants that were homozygous for the nonsense codon ($KSR1^{272*/*}$) and mutants that were homozygous for the entire B allele (*KSR1B/B*, which is homozygous for both aa189R/R and aa272*/*) (Fig 4A). We were able to recover both mutants without any additional LOH events (S11 Fig), and found that both $KSR1^{272*/*}$ and *KSR1B/B* had increased $MIC_{50}$ and SMG

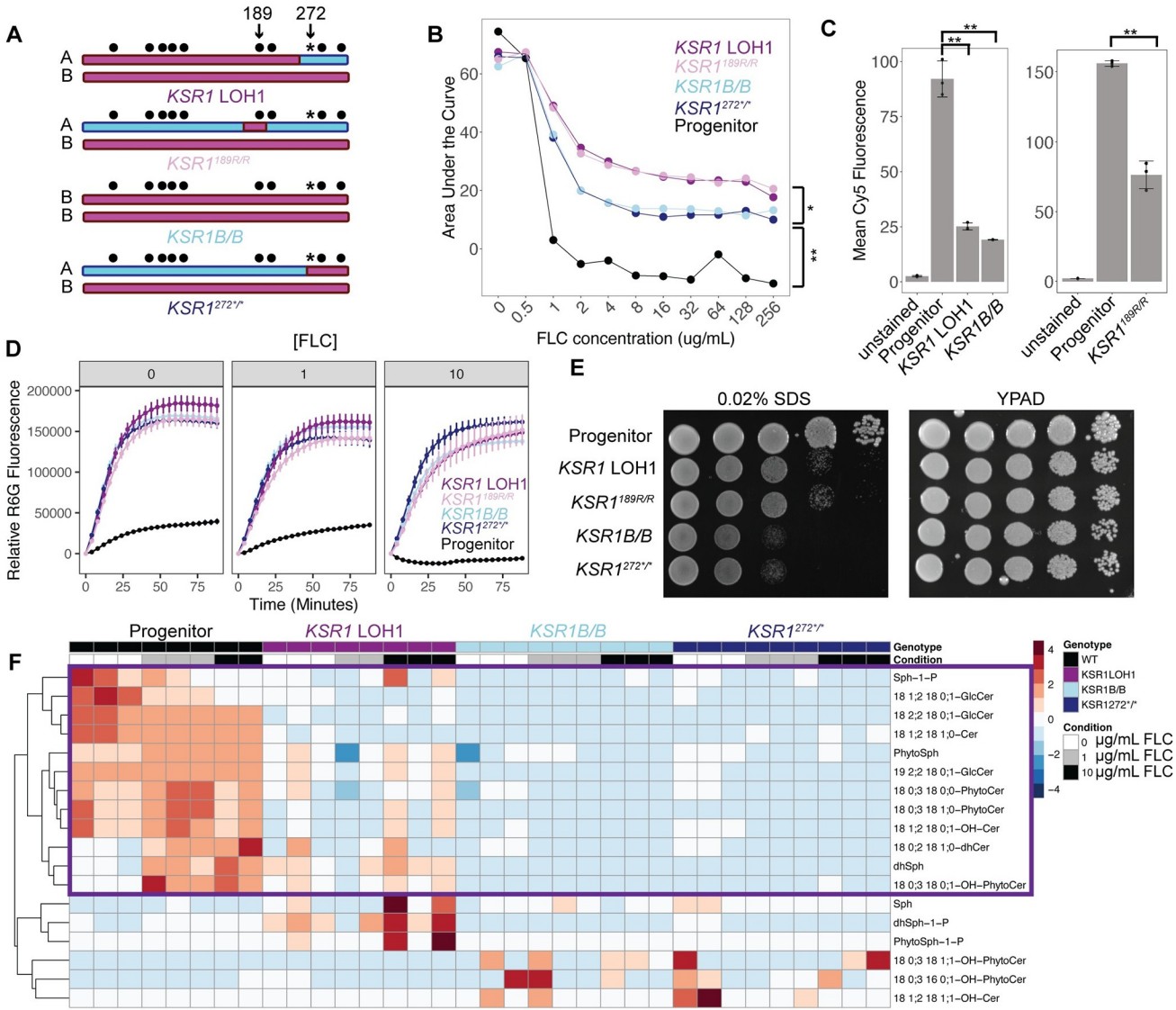

**Fig 4. LOH unlinks 189R/R from a stop codon with masking effects on phenotype.** (A) Schematics of the *KSR1* locus in four engineered strains: the strain engineered to contain the *KSR1* LOH1 (as in Fig 1J), one that is homozygous for arginine at aa189 only (*KSR1^189R/R^*), one that is completely homozygous for the entire *KSR1B* allele (*KSR1B/B*), and one that is homozygous for the nonsense codon to the end of the protein (*KSR1^272*/*^*). SNP positions are indicated by circles, and the SNP that codes for the nonsense codon in allele B is indicated by an asterisk. A and B alleles are indicated in blue and pink, respectively. (B) Area under the growth curve, calculated from 48 h liquid growth assays, is plotted on the y-axis, relative to the growth condition of various FLC concentrations shown on the x-axis. Each point is the mean of three replicates. Each strain is shown in a different color and connected by line segments. (*Indicates paired t-test values < 0.1, ** indicates paired t-test values < 0.01 for all comparisons in 2ug/mL FLC and higher) (C) Mean intracellular Cy5 fluorescence, as measured by flow cytometry, is plotted with both points and bars for each strain listed, after 4 hours of incubation with FLC-Cy5. Each point represents one replicate, bars are means and whiskers are quartiles. The same data as shown in Fig 2B is shown with the addition of the *KSR1B/B* strain for comparison, and separate panels are separate experiments (**Indicates a paired t-test p-value of < 0.01). (D) Extracellular R6G fluorescence is plotted on the y-axis over time after the addition of glucose, measured in minutes, on the x-axis. The same data as shown in Fig 2D is shown with the addition of a *KSR1^189R/R^* (pink), *KSR1B/B* (light blue), and *KSR1^272*/*^* (dark blue) strains for comparison. (E) Spot plates showing serial dilutions of strain across the x-axis at concentrations of $10^7$, $10^6$, $10^5$, $10^4$, and $10^3$ cells/mL in rich media (YPAD), and 0.02% sodium dodecyl sulfate (SDS). One representative replicate of 3 performed is shown. (F) A heatmap showing levels of sphingolipid species present in each strain and condition relative to the mean among all samples for that lipid species. Heatmap colors indicate row-wise z-scores, red being high and blue being low. Rows are lipid species, clustered by k-means clustering as shown by the dendrograms on the left, and columns are individual samples, which are not clustered. Columns include 3 replicates each of the wild-type progenitor strain (black) grown in 0 μg/mL (white), 1 μg/mL (grey), and 10 μg/mL FLC (black), and the *KSR1* LOH1 engineered strain (purple), the *KSR1B/B* strain (light blue) and *KSR1^272*/*^* (dark blue) in the same conditions.

values ($MIC_{50}$ = 4 μg/mL, SMG = 0.86) relative to the progenitor SC5314 ($MIC_{50}$ = 0.5 μg/mL, SMG = 0.01), but the $MIC_{50}$ was not elevated to the same degree as *KSR1* LOH1 ($MIC_{50}$ = 8 μg/mL, SMG = 0.86). Furthermore, growth curve analysis across a range of FLC concentrations determined that the average growth of *KSR1B/B* and *KSR1$^{272*/*}$* was nearly identical and consistently lower than the growth of *KSR1* LOH1 and *KSR1$^{189R/R}$* strains at drug concentrations > 1 μg/mL (Fig 4B, paired t-test between *KSR1* LOH1 and *KSR1B/B* p-values < 0.1). The matched growth of *KSR1$^{272*/*}$* and *KSR1B/B* (which includes aa189R/R and aa272*/*), indicates that homozygosis of the nonsense codon affects drug susceptibility itself and masks the effect of 189R/R in strains carrying both 189R/R and 272*/*.

We next asked if homozygosis of the nonsense codon 272*/* and homozygosis of 189R/R affect drug susceptibility via similar mechanisms by measuring intracellular levels of FLC at 4 hours after drug exposure. We found that *KSR1* LOH1, *KSR1$^{189R/R}$*, and *KSR1B/B* strains accumulated less intracellular FLC relative to the progenitor strain (Fig 4C, paired t-test p-value for *KSR1B/B* = 0.004, for *KSR1$^{189R/R}$* = 0.004). Furthermore, *KSR1* LOH1, *KSR1$^{189R/R}$*, *KSR1$^{272*/*}$*, and *KSR1B/B* all had increased active efflux of R6G when exposed to 0, 1, or 10 μg/mL FLC (Fig 4D) and increased susceptibility to Amphotericin B (S8 Fig). All four mutants also had a heightened sensitivity to the detergent SDS, which destabilizes cell membranes (Fig 4E). Interestingly, the strains with a homozygous nonsense codon (*KSR1B/B* and *KSR1$^{272*/*}$*) had a stronger growth defect in the presence of SDS relative to the *KSR1* LOH1 or *KSR1$^{189R/R}$* strains (Fig 4E). Thus, all mutants had similar intracellular FLC levels and efflux levels, but those with a homozygous nonsense codon were more sensitive to SDS, suggestive of a more severe membrane defect.

We next asked about differences in the sphingolipid composition of *KSR1* LOH1 relative to strains homozygous for the nonsense codon. By comparing lipidomic data for the *KSR1* LOH1, *KSR1B/B*, and *KSR1$^{272*/*}$* strains and the progenitor, we found that strains homozygous for the nonsense mutation had lower levels of most sphingolipid species relative to both the progenitor and the *KSR1* LOH1 strain (Fig 4F, purple box). In addition, strains homozygous for the nonsense mutation did not have the increase in dhSph-1-p or PhytoSph-1-p that was seen in the *KSR1* LOH1 strain relative to the progenitor at 10 μg/mL FLC (Fig 4F). Together, these results suggest that both *KSR1* LOH1 and strains homozygous for the nonsense codon have decreased drug susceptibility related to increased drug efflux pump activity likely due to changes in their sphingolipid composition, and that changes in sphingolipid levels are more severe in strains homozygous for the nonsense codon.

The SDS sensitivity of the strains prompted us to examine their vacuole morphology, which can be an indication of membrane stress [37]. Indeed, using FM4-64 to stain vacuolar membranes, we observed some vacuolar abnormality: in FLC, both *KSR1* LOH1 and *KSR1B/B* strains had smaller vacuolar area than the progenitor strain (S12A and S12B Fig), yet cells were similar in size and morphology to the progenitor strain (S12A Fig).

Lipid droplets affect cellular stress responses by sequestering excess toxic lipids and sterols and mediating the release and transport of lipids from the ER to the plasma membrane and other cellular organelles [61]. Using BODIPY staining to visualize lipid droplets, we found that the *KSR1* LOH1 strain had, on average, a higher, but not significantly different, number of lipid droplets per cell as compared to the progenitor, in rich media. Conversely, the *KSR1B/B* strain had, on average, a significantly lower number of BODIPY stained lipid droplets, which was similar to the decrease in the number of lipid droplets seen when wild-type cells were treated with myriocin (S13A and S13B Fig). Since myriocin inhibits the step in sphingolipid biosynthesis immediately preceding the reaction catalyzed by *KSR1* [35,36], this indicates that *KSR1B/B* shares a phenotype characteristic of sphingolipid pathway inhibition while the *KSR1* LOH1 does not. Together, these results indicate that *KSR1* LOH1 as well as strains with

homozygous nonsense codon affect lipid homeostasis, membrane stress, and the formation of lipid droplets, but that these two types of mutations differ in their effect on lipid droplet properties.

### Nonsense codon at aa272 results in mislocalization of Ksr1p

The *KSR1* nonsense codon is predicted to truncate the protein immediately upstream of a putative membrane-anchoring domain encoded by *KSR1A* in *C. albicans* and in *TSC10*, the *S. cerevisiae* ortholog (Fig 5A, [39,62,63]). To ask if this C-terminal tail affects Ksr1p localization, we fused an mNeon-Green fluorescent protein to the N-terminus of either *KSR1A* or *KSR1B* in the SC5314 background. As expected based on the localization of Tsc10p in *S. cerevisiae*, mNeonGreen-Ksr1A co-localized with an ER stain (Fig 5B, [64]). However, mNeonGreen-

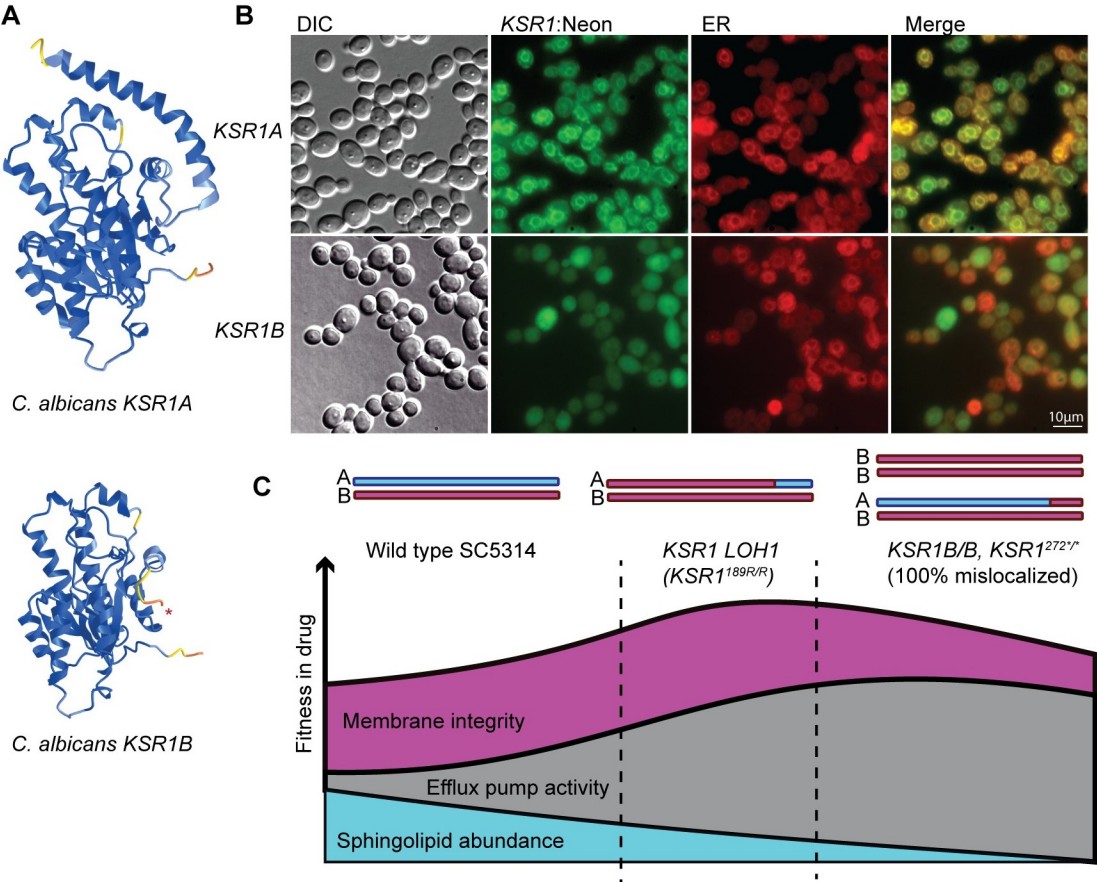

**Fig 5. Nonsense allele results in mislocalization of Ksr1p.** (A) Ribbon models produced by Alphafold [62] predictions are shown for *C. albicans* Ksr1pA and Ksr1pB. The early stop codon in Ksr1pB (red star) truncates the protein ahead of a putative membrane-anchoring domain positioned on the right side of the model. Models are colored according to Alphafold confidence scores, in which blue is high and yellow to red are low. (B) Fluorescence microscopy images for engineered strains containing *KSR1A* (top row) or *KSR1B* (bottom row) tagged with NeonGreen. From left to right the columns show DIC, Neon green fluorescence, the endoplasmic reticulum-specific probe 2 [64], and a merged image of mNeonGreen and ER probe. (C) A model reflecting the levels of sphingolipid abundance, rate of active efflux, and membrane sensitivity together contributing to a theoretical fitness landscape. The regions on this landscape occupied by a completely wild-type Ksr1p (left), the evolved LOH strains that bear alleles homozygous for *KSR1*[189R/R] (middle), and strains with a homozygous nonsense mutation resulting in mislocalization (right) are shown. The evolved LOHs occupy a 'sweet spot' with increased active efflux, intermediate sphingolipid levels, and membrane sensitivity leading to the lowest drug susceptibility and highest tolerance.

Ksr1B labeling was diffuse throughout the cytoplasm (Fig 5B). We posit that the failure of the truncated Ksr1Bp to localize to the ER likely prevents efficient contact of Ksr1p with its substrate, which is in the ER, and thus effectively reduces Ksr1Bp activity. The reduction in activity due to mislocalization appears more severe than that resulting from the homozygosis of arginine at aa189, based on the further reduction in sphingolipid species abundance in strains with homozygous nonsense alleles (Fig 4F). However, rather than further increasing drug efflux pump activity, efflux in these strains remained steady (Fig 4C and 4D), and sensitivity to SDS was increased (Fig 4E). Together, these data suggest that cumulative positive and negative fitness trade-offs result in a 'sweet spot' that is occupied by the evolved strains with $KSR1^{189R/R}$ but not in those harboring homozygous nonsense codons (Fig 5C). This underscores the observation that, at the end of the evolution experiments, all four LOH events homozygosed $KSR1$ aa189 but not aa272.

## Discussion

Here we followed the evolutionary progression of a lineage of *C. albicans* adapting to the azole drug fluconazole via the sequential acquisition of CNV followed by LOH on a different chromosome occurring in the same background. Each of these genome changes affected many genes, yet we found that most of the reduced susceptibility can be attributed to changes in two genes: amplification of *NCP1* within the Chr4 CNV and partial homozygosis of *KSR1* within the ChrR LOH. These two genes were not previously associated with drug resistance and it appears that they impact drug susceptibility via different mechanisms. *NCP1* encodes a cytochrome p450 reductase that, when overexpressed, increases the FLC $MIC_{50}$ by 2-fold. Ncp1p was a likely contributor to decreased drug susceptibility because it interacts with Erg11p, the target of fluconazole. In addition, a screen for haploinsufficient increases in drug sensitivity did identify an *NCP1* heterozygous null as a sensitizer to FLC [65]. However, we are not aware of any previous descriptions of *NCP1* copy number changes conferring decreased FLC susceptibility. The Chr4 CNV was followed by a ChrR LOH event that homozygosed part of *KSR1*, which is required for proper sphingolipid biosynthesis. The homozygosis of codon 189 to arginine can explain a 16-fold increase in the $MIC_{50}$ as well as a high level of drug tolerance. When in combination with the Chr4 CNV, these two mutations are associated with an $MIC_{50}$ of >256 µg/mL FLC. Interestingly, the combination of an overexpression of *NCP1* and LOH at *KSR1* does not phenocopy the combination of the Chr4 CNV and LOH at *KSR1*, indicating other genes located in the Chr4 CNV or LOH that might also contribute to drug resistance or interact with the LOH at *KSR1*.

LOH of *KSR1* arose independently in four isolates that were selected for increased growth in fluconazole. Recurrent isolation of similar LOH breakpoints points to the potential importance of Ksr1p and sphingolipid metabolism in drug responses. All four independently evolved *KSR1* mutants were defective in filamentation, a phenotype also seen previously with *ksr1Δ/Δ* mutants [39]. We posit that the filamentation defect in *KSR1* mutations might be due to alterations in the levels of sphingolipids and ergosterol that perturb lipid raft formation and cell membrane polarization, processes critical to filamentous growth [24,66]. The filamentation defect might also lead to a decrease in virulence, which could help explain why *KSR1* mutants have not been previously identified in azole resistant clinical isolates. Indeed, in an analysis of 270 isolates with published whole genome sequencing data, arginine at position 189 and a nonsense codon at position 272 were restricted to SC5314-derived isolates (S3 Table). However, the biosynthesis of sphingolipids, and specifically glucosyl ceramides, is also required for *C. albicans* virulence independently of the yeast-to-hyphal transition [67]. The precise connection between sphingolipids, filamentation, and virulence remains to be determined [26].

One remarkable feature of the four independent LOH events observed at *KSR1* is that they all effectively unlinked two loci: codon 189, a residue important for reduced susceptibility, was homozygosed, while codon 272, which encodes a premature stop codon on the B allele, always remained heterozygous. The power of LOH to act as a type of allelic recombination might help explain the lack of a predicted 'genetic breakdown,' also known as Muller's ratchet [68], observed in asexual diploid species, in which stop codons or loss of function mutations are allowed to accumulate with deleterious effects [42].

The nonsense allele at codon 272 truncates Ksr1p just prior to its localization domain and consequently results in Ksr1p mislocalization. The mislocalization of Ksr1p likely inhibits its ability to interact with its substrate in the ER, greatly diminishing enzyme efficiency. However, the nonsense mutation in *KSR1B* does not appear to be a null allele, based on our ability to obtain *KSR1B/B* transformants. We propose that, as in other fungal species [56], Ksr1p forms a homodimer. In a strain heterozygous for the nonsense codon at *KSR1*, localization of Ksr1p would be only partially reduced, because heterodimers would be able to localize partially, while homodimers would fail to localize properly. Thus, we propose that complete Ksr1p mis-localization severely reduces sphingolipid levels, affecting the cell membrane structure and resulting in an increase in active drug efflux (Fig 5C). By contrast, 189R/R homozygosity local-izes to the ER and only partially inhibits sphingolipid biosynthesis. Thus, 189R/R homozygosis results in a 'sweet spot' for drug susceptibility, in which efflux is highly active, there is a lower accumulation of the toxic dienol, and the membrane is not as severely compromised (Fig 5C).

*KSR1* Arg189 is predicted to lie at the interface between the two subunits of the dimer in *Cryptococcus neoformans*, where the structure of the homodimer was recently solved [56], and it also is in close proximity to the well-conserved catalytic triad [39]. Therefore, homozygosity of arginine at amino acid 189 could influence dimerization, enzymatic specificity and/or enzyme activity. Based on the observed reduction in sphingolipid species, 189R/R appears to reduce Ksr1p function relative to the progenitor that is heterozygous (189R/Q). Interestingly, in wild-type cells, inhibition of sphingolipid biosynthesis with myriocin, NPD827 or aureoba-sidin A, increases FLC sensitivity rather than decreasing it [35–37]; similarly, other loss of function mutations in sphingolipid genes tend to increase, rather than decrease FLC suscepti-bility [26,34]. This does not appear to be the case for *KSR1* and suggests that the relationship between drug susceptibility and sphingolipid abundance phenotypes may be more complex than previously appreciated. Indeed, although connections have been made between sphingo-lipid abundance, the activity of drug efflux pumps, and the presence of the toxic dienol, exactly how these are connected remains to be determined.

In summary, here we observed an increase in tolerance early in the evolution experiment that was associated with the early appearance of CNV and aneuploidy, followed by the acquisi-tion of a short LOH event that resulted in strong azole resistance. Both CNV and LOH occur at higher frequencies than de-novo point mutations [5–8], and therefore the co-occurrence of these types of mutations in the same background is likely. The LOHs we observe here are short, much like gene conversion events, and could therefore easily be overlooked in genome sequencing. Indeed, despite their importance, identifying CNV and LOH remains challenging, particularly in whole-population sequencing. In addition, because they often impact multiple genes, determining the specific genes and mechanisms by which CNV and LOH impact FLC sensitivity is difficult. This study demonstrates that the rapid acquisition of large genome changes and LOH events are frequent enough to co-occur and to rapidly acquire antifungal drug resistance. This work sheds light on genes that contribute to this process, and further studies elucidating the mechanisms of drug resistance behind other CNV and LOH events will continue to improve our understanding of the forces at play during adaptation to antifungal drugs.

## Materials and methods

### Ethics statement

The mouse studies were approved by the Institutional Animal Care and Use Committee at the Lundquist Institute for Biomedical Innovation at Harbor-UCLA Medical Center.

### Microbroth dilution assays ($MIC_{50}$ and SMG) and growth curve assays

Strains were grown from frozen glycerol stocks overnight in 3 mL of YPAD containing 2% dextrose (20 g/L Bactopeptone, 10 g/L yeast extract, 0.04 g/L adenine, 0.08 g/L uridine). Strains were then diluted to a final calculated OD of 0.01 in YPAD containing 1% dextrose and added to a 96-well assay plate at a 1:10 ratio, resulting in a final starting OD of 0.001 in each well. Assay plates were prepared by performing serial dilutions of YPAD containing 1% dextrose and 256 μg/mL fluconazole, resulting in final concentrations of 256, 128, 64, 32, 16, 8, 4, 2, 1, or 0.5 μg/mL FLC with cells added, as well as YPAD with 1% dextrose without the addition of FLC.

For $MIC_{50}$ and SMG assays, 96-well plates were allowed to incubate stationary in a humidified chamber at 30˚C for 24 hours, at which time they were resuspended by pipetting and $OD_{600}$ measured using a BioTek Epoch 2 microplate reader. Plates were then allowed to sit stationary for another 24 hours (48 total) at 30˚C in a humidified chamber, after which they were resuspended and OD measured. $MIC_{50}$ was calculated by first subtracting blank values from all wells and then normalizing each strain grown in FLC to the same strain grown in YPAD without FLC and then averaged across replicates. The drug concentration at which average growth was reduced below 50% at 24 hours was designated as the $MIC_{50}$. SMG was calculated by averaging the normalized growth at 48 hours for all concentrations of FLC greater than the $MIC_{50}$ value. The triplicate measurements were averaged to produce the final reported SMG value.

For growth curves, 96-well plates containing sterilized water motes surrounding the outer wells were prepared as described above for microbroth dilution assays and grown in a BioTek Epoch 2 at 30˚C with shaking for 48 hours, with $OD_{600}$ readings taken every 15 minutes. Data was analyzed in R (version 4.1.3) using the package growthcurver [69] to calculate empirical AUC values. Mean and SE of triplicate $OD_{600}$ values (minus blanks) were calculated and plotted over time using ggplot2 for all growth curve plots.

### Evolution experiments

The lineage described in Fig 1 was evolved as previously described in [49]. As detailed there, a single colony isolated from susceptible lab strain SC5314 was suspended in 1 mL sterile YPAD media and grown overnight at 30˚C. One fourth of this culture was used to initiate a lineage grown in YPAD + 1 μg/mL FLC in a deep-well 96-well plate which was sealed with Breathe EASIER tape and placed in a humidified chamber at 30˚C for 72 hours. Every 72 hours, cells were resuspended and transferred to a new plate with fresh YPAD + 1 μg/mL FLC at a dilution of 1:1000 for a total of 10 transfers. Culture remaining from each passage was frozen at -80˚C in 20% glycerol.

In this second experiment, *C. albicans* cells were evolved in liquid YPD (2% bacto-peptone, 1% yeast extract, 2% dextrose (filter-sterilized)) + 1 μg/mL FLC. The experiment consisted of 4 lineages each initiated from a single colony SC5314 progenitor and passaged at a 1:100 dilution every 24 hours for a total of 12 passages (30˚C, 200 rpm), approximating 80 generations. Every 4 days, the passaged cultures were frozen at -80˚C in 25% glycerol. The final passage of each lineage was frozen at -80˚C in 25% glycerol and selected for further analysis.

## Whole genome sequencing, variant detection, and allele frequency analysis

To ensure that the results of whole genome sequencing for evolution passages reflected the entire populations as much as possible, each glycerol stocked passage was thawed and half the entire volume used for a short (~16 hours) growth expansion in 0.5 μg/mL FLC before collection of genomic DNA for sequencing. For all other whole genome sequencing experiments, strains were grown in 3 mL YPAD containing 2% dextrose overnight at 30°C with shaking. Cells were pelleted and resuspended in TENTS buffer (1% SDS, 100 mM NaCl, 100 mM Tris pH 8, 1 mM EDTA, 2% Triton), added to ~250 uL of .5 mm glass beads, and lysed in a BeadRuptor Elite (Omni International, 1 cycle, 4 m/s, 15 s). Genomic DNA was then isolated using a phenol-chloroform extraction. Sequencing libraries were prepared by SeqCenter, LLC, or SeqCoast Genomics using the Illumina DNA Prep and sequenced on an Illumina NovaSeq 6000. Adaptor sequences and low-quality reads were removed using Trimmomatic (v0.39, parameters LEADING:3 TRAILING:3 SLIDINGWINDOW:4:15 MINLEN:36 TOPHRED33) [70]. Trimmed reads were mapped to the *C. albicans* reference genome (A21-s02-m09-r08) from the Candida Genome Database (CGD) (http://www.candidagenome.org/download/sequence/C_albicans_SC5314/Assembly21/). Reads were mapped using BWA-MEM (v0.7.17) with default parameters [71]. Samtools (v1.10) [72] was used to sort, index, and remove PCR duplicates from aligned reads, as well as calculate read depth at each genomic nucleotide position using the 'samtools depth' function. Single nucleotide variants were called using GATK (v4.1.2) [73] Mutect2, with SC5314 designated as the 'normal' sample to identify new SNVs, or with the mutant designated as the 'normal' sample to identify regions of LOH, using default parameters. Variants differing between evolved and progenitor strains were then manually inspected using the Integrated Genome Viewer (IGV, v2.12.3) [74]. Biallelic SNPs for allele frequencies were identified using GATK (v4.1.2) 'HaplotypeCaller' [75] followed by 'GenotypeGVCFs', both set with -ploidy 2, and then selected using 'SelectVariants' with parameters -select-type SNP and–restrict-alleles-to BIALLELIC. Variants were then filtered for quality using QD < 2.0, QUAL < 30.0, SOR > 3.0, FS > 60.0, MQ < 40.0, MQRankSum < -12.5, and ReadPosRankSum < -8.0.

Single nucleotide polymorphisms in the *KSR1* gene of 270 whole genome-sequenced *C. albicans* isolates (S3 Table) were retrieved as fasta files and compared using the MUSCLE Mutliple sequence alignment tool [76]. The impact on the Ksr1 protein sequence was evaluated across isolates using the multiple sequence alignment editor SeaView [77]. All sequences were retrieved from the NCBI Genomes database [78–82].

## CNV analysis

Whole genome sequencing was processed as described above, and sequencing depth files generated using 'samtools depth' were downloaded and further processed in R (v4.1.3) to identify regions of varying copy number. Depth files were read into R, and rolling means were calculated for 500 bp windows across the entire genome using the RcppRoll R package (v0.3.0) [83]. Means were then divided by the median read depth of the entire genome (excluding mitochondrial DNA) to calculate relative abundance and then multiplied by 2 (due to *C. albicans* being a diploid organism) to generate an estimated copy number. Estimated copy numbers were then plotted according to genomic position to visualize regions of the genome with changes in copy number. Copy number based on read depth and heterozygosity of the entire genome was also visualized using the online YMAP platform [84]. Raw fastq files were uploaded to YMAP and processed using the parameters "ploidy of experiment: 2, baseline ploidy:2, generate Figs with annotations? Yes, data type: whole genome NGS (short reads), read type: paired-end short reads, reference genome: Candida albicans SC5314 (CGD:

A21-s02-m09-r10), haplotype map: Abbey_et_al_2014, chromosome-end bias (forces using GC content bias) checked."

For mixed populations, such as the entire passage populations of the first evolution experiment, abundance of trisomies or CNVs in the population was estimated using the copy numbers calculated for individual clones isolated from the populations as a reference. The formula $A^*x + B^*(1-x) = C$, where B is euploid copy number, A is the CNV copy number estimated from a single colony, and C is the estimated copy number from the entire population, was used to find x, the proportion of the population containing the CNV, assuming no other CNV of a different copy number is present in the same region.

## Strain engineering

All primers and sequences used in strain construction and verification are listed in S2 Table. All strains described in this paper (S1 Table) were whole genome sequenced, and the whole genomes sequencing data can be found at the NCBI Sequence Read Archive (BioProject ID PRJNA1071177).

## Construction of *KSR1* LOH1, *KSR1*^272*/*^, *KSR1B/B*, and *KSR1A/A* strains

*KSR1* LOH engineered strains, including *KSR1* LOH1, *KSR1*^272*/*^ and *KSR1B/B* were constructed using homologous recombination. Transformation constructs were generated by amplifying three fragments that were subsequently stitched together into one fragment using strand overlap extension PCR: 1) a 398 bp fragment downstream of the *KSR1* locus (relative to the Crick strand) amplified from genomic DNA of the evolved strain containing the *KSR1* LOH, 2) a NAT resistance marker optimized for *C. albicans* from plasmid pAS3019, and 3) the entire *KSR1* coding sequence and 464bp immediately upstream (relative to the Crick strand) amplified from genomic DNA of the evolved strain containing the *KSR1* locus. Amplification of the *KSR1* locus to generate the transformation construct was not allele-specific, and therefore a mixed pool of transformation constructs was generated that included both *KSR1B* and the chimeric combination of *KSR1A* and *KSR1B* that resulted from LOH1. Transformation was performed in SC5314 by lithium acetate and heat shock, and single colonies isolated. Single colonies were screened by PCR for correct upstream and downstream insertion into the genome. Sanger sequencing was then performed for the entire *KSR1* locus specifically for the transformed allele by amplifying fragments from within the NAT resistance marker to determine the combination of A and B SNVs present. The genotype of the non-transformed allele was determined by amplification the *KSR1* locus, selection of the smaller band (excluding NAT), and restriction digest with restriction enzyme BanII, which will cut the A allele sequence at nucleotide 192 (GAGCCC) but not the B allele sequence (GAACCC). In this way, three strains were selected: 1) one in which the entire A allele had been transformed with the new chimeric allele, resulting in the identical genotype to the evolved strain at the *KSR1* locus, 2) one in which the A allele had been partially transformed with the B allele construct, resulting in *KSR*^272*/*^ that was otherwise heterozygous, and 3) one in which the entire A allele had been transformed into a complete B allele at the *KSR1* locus resulting in *KSR1B/B*. Whole genome sequencing was then performed to ensure the correct genotypes at *KSR1* and ensure that these strains did not contain additional changes such as aneuploidies or other LOH. The same process was used to generate the *KSR1A/A* strain, using genomic DNA from the wild-type SC5314 to amplify the *KSR1A* locus.

## Construction of *tetO-NCP1* strains

Strains containing an tetO-*NCP1* overexpression construct were generated by first amplifying a transformation construct using primers with homology to the native *NCP1* locus from a tetO vector pAS3027. The transformation construct, including upstream and downstream homology, included CaTar-TetO-FlpNAT. This construct was transformed into lab strain Sn152, an auxotrophic derivative of SC5314, by lithium acetate and heat shock transformation. Transformants were screened for correct insertion using PCR for both upstream and downstream flanks of the transformed region. Sanger sequencing was used to determine which allele of *NCP1* was overexpressed. For strains containing both the tetO-*NCP1* and *KSR1* LOH1 constructs, NAT was removed from the tetO-*NCP1* strains by growth on FBS to activate FLP recombinase, and then transformed with the same protocol described above for *KSR1* LOH1.

## Construction of *KSR1* deletion mutants in SC5314

The *KSR1* deletion mutants in the SC5314 background were constructed using the CRISPR system developed by [85]. The intact CRISPR cassette was built with plasmids pADH118 for gRNA expression and plasmid pADH99 for Cas9 components. Both plasmids had the carbenicillin selection marker, and when cut and stitched together they reconstituted the NAT marker. The LEUpOUT strategy was used to remove the CRISPR components and the NAT marker. The *C. albicans* strain AHY940 (SC5314 *LEU2* heterozygous knockout) was used to transform and produce the mutants. The parental strain and plasmids were provided by the Hernday lab.

## Construction of strains carrying *mNeon-Green-Ksr1p*

To generate *C. albicans* strains with a tagged Ksr1 protein, we amplified: *NAT1* from plasmid pJB-T155 (primers 2313 and 2251), mNeonGreen from plasmid pJB-T409 (primers 2315 and 2369), the plasmid backbone containing an *E. coli* origin of replication, and the Amp$^R$ gene from pJB-T155 (primers 2325 and 2027). The promoter sequence of *TDH3* was amplified from the genomic DNA of *C. albicans* SC5314 (primers 2239 and 2314).

The plasmid expression cassette (*NAT1-TDH3-mNeonGreen*) was amplified from pJB-T510 by PCR using PCRBIO VeriFi Mix with primers CB-KSR1-F, CB-KSR1-R, which are complementary to 40 bp 5'-to the start codon of the *KSR1* ORF, and 40 bp downstream of the start codon. The amplified PCR product was used to transform a fresh *C. albicans* SC5314 log phase liquid culture and transformants were selected on YPD + nourseothricin (400 μg/ml).

Successful transformants were validated by diagnostic PCR using PhireGreen Hot Start II PCR Master Mix along with primers specific for the expression cassette and *KSR1* (#2039, #CB-KSR1-CHK). To determine which allele of the gene was fused to mNeonGreen, the sequence was amplified by PCR (primers #2039, #2342), cleaned using a Qiagen PCR cleaning kit, and Sanger sequenced.

## Construction of *KSR1*$^{189R/R}$

The *KSR1*$^{189R/R}$ strain was constructed by homologous recombination of a mutated amplicon (generated by oligo-directed mutagenesis) marked with NAT into the progenitor SC5314 background. The mutated amplicon was generated by oligo-directed mutagenesis. Specifically, the second half of the *KSR1A* allele and the NAT selection marker were amplified from the *KSR1A/A* strain background using primer 1624 and primer 2042. Primer 2042 contains the nucleotide change *KSR1*:A566G which codes for amino acid arginine at position 189 in the *KSR1B* allele. This construct was transformed into SC5314 using heat shock and transformants

were selected on nourseothricin plates. Single colony transformants were then screened for reduced filamentation at 37C to select for colonies in which the A allele had been modified. Colonies with reduced filamentation were then screened for correct insertion of the construct by PCR using primers 1660, 1661, 1659, and 1154 (S2 Table) and amplicons Sanger sequenced. Transformants with correct construct sequences were then whole genome sequenced to verify the correct sequences of *KSR1A* and *KSR1B* alleles. Strain AMS6412 was selected due to no additional LOH or aneuploidies detected by whole genome sequencing. MIC assays performed on this strain were performed with a different FLC lot than previous MIC assays. The MIC/SMG exactly matched the MIC of the *KSR1* LOH1 strain when done in parallel with the same FLC lot. However, the MICs of both strains and controls were lower than the assays done with the previous lot, with an MIC value of 2.

## Alphafold protein structure predictions

Ribbon models of structural predictions for *KSR1A* and *KSR1B* were generated using *Alphafold* (version 2.3.2) [62]. Protein coding sequences for *KSR1A* and *KSR1B* were downloaded from the Candida Genome Database [86] from the V22 reference genome and submitted for Alphafold predictions using the colab notebook found at: https://colab.research.google.com/github/deepmind/alphafold/blob/main/notebooks/AlphaFold.ipynb, using default parameters.

## Fluorescence microscopy of tagged *KSR1* alleles

Cells were picked from an agar plate into SDC medium and grown overnight by incubating at 30˚C with shaking. The following day, cells were diluted 1:50 into fresh SDC. For fluorescence microscopy of the mNeonGreen-tagged protein and staining with an endoplasmic reticulum-specific probe 2 [64] the cells were incubated at 30˚C for 2 hours with shaking, the stain was added to a final concentration of 1 μM, and the cells were incubated for an additional hour in the dark. 3–5μL of cell culture was deposited on a glass slide topped with 10 μL of low melt agar and imaged with a Nikon Ti Eclipse microscope using the Nikon Elements AR software, and Nikon Plan Apo x100 objective. Filters used were: 488 nm mNeonGreen excitation and 510 nm for emission; for the ER-specific stain CFP 438 nm excitation and 483 nm emission.

## Lipid analysis

Strains AMS2401, AMS5778, AMS5779, AMS5784, AMS5780, and AMS5782 were grown overnight in 10 ml of SDC at 30˚C under agitation. Cell suspensions were centrifuged, washed in sterile PBS and counted. Then, $1x10^8$ cells were inoculated in 10 ml of fresh SDC only, or in media containing 1 or 10 μg/ml of fluconazole and grown for additional 48 h at 30˚C. Prior to cell lysis, C17-sphingolipid standards were added to the samples [87,88]. Mandala extraction was carried out as described previously [89], followed by Bligh and Dyer Extraction [90]. A third of each sample obtained from the Bligh and Dyer Extraction was reserved for inorganic phosphate (Pi) determination, so the relative sphingolipid signal was normalized by the Pi abundance. The organic phase was transferred to a new tube and submitted to alkaline hydrolysis of phospholipids [91]. Finally, the organic phase was dried and used for mass spectrometry analysis [88]. Sphingolipid abundances, normalized by Pi abundance, were then analyzed using R (version 4.1.3). Any lipid species that were not detected in any sample were removed from analysis. Because the distribution of lipid abundances was exponential across samples, $\log_{10}$ values were calculated for all normalized abundances, and heatmaps generated using the $\log_{10}$ transformed values. Heatmaps were generated using the R package pheatmap (version 1.0.12) [92], and scaled by row (across samples) to result in heatmap color intensities that represent each sample's z-score calculated from the distribution of all samples in the row. Two

samples that clustered away from all other samples with a cluster dendrogram height of >8 were removed as outliers.

## RNA sequencing

*C. albicans* cells were cultured overnight in liquid YPD medium (2% bacto-peptone, 1% yeast extract, 2% dextrose (filter-sterilized)) at 30˚C with shaking (200 rpm) and then diluted 1/50 in YPD either without FLC, with 2 μg/ml FLC, or with 64 μg/ml FLC. After 24 h at 30˚C with shaking (200 rpm), 1 ml of the cell suspension was utilized to extract total RNA using the Ribo-pure-Yeast RNA kit (AM1926, Invitrogen), according to the manufacturer's instructions. A Bioanalyzer (Agilent) was used for the qualitative sample analysis, and only those with RNA quality (RIN) scores of 7 or higher and a 260/280 ratio within the range of 2.13–2.2 were sequenced. Sequencing was performed with the help of the Biomics Platform at Institut Pasteur using pair-end Illumina stranded-mRNA sequencing. RNA sequences reported in this paper have been deposited in the NCBI Sequence Read Archive, https://www.ncbi.nlm.nih.gov/bioproject (BioProject ID PRJNA1063495).

## RNA sequencing data analysis

The paired reads from the RNA-seq libraries were trimmed for low-quality reads, and Illumina TruSeq adapters were removed with Cutadapt v1.9.1 [93] with the following parameters:—trim-qualities 30 -e (maximum error rate) 0.1—times 3—overlap 6—minimum-length 30. The cleaning of rRNA sequences was performed with Bowtie2 v2.3.3 [94] with default parameters. The cleaned reads from RNA-seq paired-end libraries were aligned to the *C. albicans* SC5314 reference genome (Version_A22-s07-m01) with Tophat2 v2.0.14 [95] and the following parameters: minimum intron length 30; minimum intron coverage 30; minimum intron segment 30; maximum intron length 4000; maximum multihits 1; and microexon search. Genes were counted using featureCounts v2.0.0 [96] from the Subreads package (parameters: -t gene -g gene_id -s 2 -p). Analysis of differential expression data was performed in DeSeq2 [97]. The statistical analysis process includes data normalization, graphical exploration of raw and normalized data, tests for differential expression for each feature between the conditions, and raw P-value adjustment. The genes with adjusted P-value less than 0.1 were considered as differentially expressed compared to the non-treated condition (YPD no FLC). $Log_2$ fold changes for each evolved strain relative to the progenitor strain in no FLC were plotted as heatmaps using *pheatmap* [92] without any additional scaling, and asterisks were used to denote significant P-values as described above.

## Cy5 and R6G flow cytometry

*C. albicans* cells were grown overnight in liquid YPD medium (1% yeast extract, 2% peptone, and 2% dextrose (with the dextrose filter sterilized)) at 30˚C with shaking (200 rpm). Cultures were diluted 1 in 100 in a 3 mL volume of YPD and compounds were added to final concentrations of 1 μg/mL for Rhodamine 6G (Sigma), or 1 μM for FLC-Cy5. Samples incubated with 10 μg/mL of Beauvericin (Sigma) were used as positive controls and those incubated with corresponding volumes of DMSO were used as negative controls. Cells were incubated at 30˚C with shaking and harvested after 4 h, washed twice with sterile PBS and diluted to a density of ~$10^6$ cells/mL in PBS. Fluorescence was measured in 200 μL aliquots of cell suspension and data were collected from 100,000 cells per sample using a Miltenyi MACSQuant Analyzer 10 Flow Cytometer. Cell populations were gated by SSC/FSC to eliminate small debris particles. Experiments were performed with at least three biological and two technical replicates. Analyses were performed using FlowJo 10.8.

## Rhodamine 6G efflux assay

*C. albicans* cells were patched from glycerol stocks onto YPAD agar plates and grown at 30˚C for ~1 day. Patches were used to inoculate 3 mL liquid YPAD media with 2% dextrose (20 g/L bactopeptone, 10 g/L yeast extract, 0.04 g/L adenine, 0.08 g/L uridine), and grown at 30˚C overnight with shaking. Cultures were then back-diluted, adding 50 μL of overnight culture into 5 mL of fresh media with 0 μg/mL FLC, 1 μg/mL FLC, or 10 μg/mL FLC added at grown for 3 hours at 30˚C with shaking. Cultures were spun down and washed with PBS (pH 7, room temperature), and then resuspended in 2 mL PBS and incubated for 2 hours at 30˚C with shaking. Rhodamine 6G (ThermoFisher #419010050) was added to a final concentration of 10 μg/mL and cultures incubated for 1 hour at 30˚C with shaking. Cultures were spun down and washed twice with 4˚C PBS, and then resuspended in a final volume of 900 μL of PBS. Cultures were then split across 6 wells of a 96-well plate to constitute technical triplicate for a set with and without dextrose (at 150 μL per well). Baseline fluorescence and optical density ($OD_{600}$) measurements were taken using a BioTrex HTX plate reader (excitation 485 nm, emission 525 nm) for 5 minutes at 1 minute intervals. The plate was then removed, and dextrose was added to the 'with dextrose' samples to a final concentration of 1%. The plate was then returned to the plate reader and fluorescence and OD measurements were taken every 4 min for a total of 90 min. Three biological replicates were performed.

## Fluorescent imaging of lipid droplets

C. albicans cells were cultured in liquid YPD medium (2% bacto-peptone, 1% yeast extract, 2% dextrose (filter-sterilized)) overnight at 30˚C with continuous shaking (200 rpm). The number of cells was determined by measuring optical densities ($OD_{600}$) using a Biotek Epoch 2 plate reader. 106 cells/mL were resuspended in YPD in different conditions: no drug, with 128 μg/ml Fluconazole (FLC, PHR1160-1G, Sigma Aldrich), or with 128 μg/ml myriocin (MYO, M1177-5M, Sigma Aldrich). The cell suspensions were incubated for 3 h at 30˚C (200 rpm). After incubation, cells were washed twice with sterile PBS and incubated with 1 μg/ml Bodipy 493/503 (D3922, Invitrogen) for 10 minutes at 30˚C and 200 rpm. The cell suspension was washed twice with sterile PBS. The cells were then imaged at a 60X magnification with a GFP filter and an exposure of 125 ms on a Zeiss AxioVision Rel. 4.8 microscope. Cellprofiler4.2.6 (https://cellprofiler.org/, available at https://github.com/CellProfiler/CellProfiler) was used to analyze the images. The Speckle Counting pipeline (https://cellprofiler.org/examples) was used to identify smaller objects within larger ones and to establish a relationship between them. Objects with a diameter of 30–100 pixels were considered as yeast cells, while those with a diameter of 1–6 pixels were classified as lipid bodies. The same pipeline was used to quantify several parameters. These include the number of lipid bodies within each cell, the highest pixel intensity within each lipid body (referred to as maximum brightness), and the integrated intensity of lipid bodies. The integrated intensity was quantified by summing the pixel intensities within these structures. All assays were performed with biological triplicates, with at least 200 cells analyzed per replicate.

## Fluorescent imaging of vacuoles

C. albicans cells were cultured in liquid YPD medium (2% bacto-peptone, 1% yeast extract, 2% dextrose (filter-sterilized)) overnight at 30˚C with shaking (200 rpm) and resuspended in YPD or YPD + FLC (128 μg/ml) at a concentration of 106 cells/mL. To stain the vacuoles, cells were incubated with 8 μM of SynaptoRed C2 (S6689, Sigma) for 3 h at 30˚C with shaking. The cell suspension was then washed with 1 mL of fresh YPD and transferred to 4 mL of YPD before incubating for 90 min at 30˚C (200 rpm). The cells were washed with sterile PBS before

imaging at a 60X magnification with an mCherry filter and an exposure of 100 ms on a Zeiss AxioVision Rel. 4.8 microscope. Cellprofiler4.2.6 (https://cellprofiler.org/, available at https://github.com/CellProfiler/CellProfiler) was used to analyze the images. The Speckle Counting pipeline (https://cellprofiler.org/examples) was used to identify objects with diameters of 10–35 pixels, these objects were considered as vacuoles. The sum of the pixel intensities within these vacuoles, represented by the integrated intensity, and the area of these vacuoles were quantified using Speckle Counting. All assays were performed with biological triplicates, with at least 200 cells analyzed for each replicate.

## RT-qPCR

*C. albicans* cells were patched onto solid YPAD agar from frozen glycerol stocks and grown at 30˚C for ~2 days. Patches were used to inoculate 3 mL YPAD media (2% dextrose) and grown overnight at 30˚C with shaking. 350 μL of overnight cultures were back diluted into 50mL fresh YPAD media with 0 μg/mL FLC or 1 μg/mL FLC added (and/or doxycycline supplemented with iron for tetO-*NCP1* strains) and grown at 30˚C with shaking until cultures reached an OD between 0.4–0.5. Each culture was pelleted, supernatant removed, and cell pellets flash frozen in liquid nitrogen, and then stored at -80˚C until RNA could be harvested. RNA was extracted using the Qiagen RNeasy Mini Kit (CAT# 74106), using the mechanical disruption method with an Omni International Bead Ruptor Elite. 1 ug of RNA from each sample was DNase treated after elution, via incubation with DNase at 37˚C for 30 min followed by the addition of EDTA to a final concentration of 5mM and heat inactivation at 75˚C for 5 min. cDNA was prepared using SuperScript II RT (ThermoFisher, Cat# 18064014) using an Oligo(dT) primer according to the manufacturer's recommendations. cDNA was used as template for qPCR reactions performed in technical triplicate with primers specific to either *TEF1*, *ACT1*, *NCP1*, or *CDR1* for each sample as well as noRT and no template controls, using PowerUp SYBR Green Mastermix (Fisher Scientific, CAT# A25742) run on a BioRad CFX-96. Delta-delta Ct values for *CDR1* were calculated, with propagation of error, relative to either *TEF1* or *ACT1* and then relative to the progenitor strain in 0 μg/mL FLC. Delta-delta Ct values for *NCP1* were calculated with propagation of error relative to *ACT1* and then relative to the progenitor strain in 0 μg/mL FLC. Three complete biological replicates were performed.

## Spot plates

*C. albicans* cells were patched onto solid YPAD agar plates and grown at 30˚C for ~2 days. Patches were used to inoculate 3 mL of liquid YPAD media (2% dextrose) and grown overnight at 30˚C with shaking. Cultures were diluted to ~$10^7$ cells/mL in PBS and serial dilutions performed to generate suspensions at $10^6$, $10^5$, $10^4$, and $10^3$ cells/mL. 5 μL of each suspension was spotted onto YPAD agar plates with 0% or 0.02% sodium dodecyl sulfate. Cells were incubated at 30˚C and pictures taken at 24 and 48 hours.

## Filamentation assays

*C. albicans* cells were cultured in liquid YPD medium (2% bacto-peptone, 1% yeast extract, 2% dextrose (filter-sterilized)) overnight at 30˚C with shaking (200 rpm). Yeast cells were diluted 1/50 in 4 ml of YPD and seeded in 6-well plates. After 2 h at 37˚C with shaking (200 rpm), the cell suspension was then washed twice with sterile PBS and imaged at a 60X magnification in brightfield on a Zeiss AxioVision Rel. 4.8 microscope. All assays were performed with three biological replicates, with at least 200 cells analyzed for each replicate. Representative images are shown in the Figs.

## *Galleria mellonella* infections

*G. mellonella* larvae were sourced from La ferme aux Coleos (Cherbourg-en-Cotentin, France), maintained at room temperature, and were used within one week of delivery. Larvae without any signs of melanization and with an average weight of 0.4 g were selected for each experiment in groups of 12. *C. albicans* cells were cultured in liquid YPD medium (2% bacto-peptone, 1% yeast extract, 2% dextrose, filter-sterilized). Following an overnight incubation at 30°C with shaking (200 rpm), cells were washed twice with sterile PBS. The cell concentrations were then determined using a Luna FX7 Automatic Cell Counter (Logos Biosystems) and adjusted to $3 \times 10^7$ cells/mL in sterile PBS. For the infection, each larva was injected with $3 \times 10^5$ cells/mL through the last left pro-leg using a 10 μL glass syringe and a 26S gauge needle (Hamilton, 80300). A subsequent injection with either fluconazole (0.2 μg/larva) or a sterile vehicle (PBS at matching injection volumes) was administered via the last right pro-leg, 2 h post-infection. The inoculum size was verified by plating fungal cells on YPD and counting the resulting colony forming units (CFUs). Both infected and control groups of larvae were maintained at 37°C for 14 days, with daily monitoring of survival. Larvae were deemed dead if no movement was observed upon contact. The experiments were conducted with three biological replicates. Control groups of larvae included those treated with PBS or FLC without fungal infection.

## Amphotericin B e-strip sensitivity tests

Cells were grown on SAB-dex agar plates at 35C for 24 hours from glycerol stocks. Cells were then resuspended in PBS and normalized to an OD of 0.1–0.15. 100μL of the normalized cell suspension was then plated on RPMI agar plates using glass beads and an Amphotericin B e-strip (Liofilchem Ref 92153, lot 031224125) placed in the center of the plate. Plates were photographed and read after 24 hours of growth at 35C in a humidified chamber.

## Sterol analysis

Cells were grown overnight in 50mL 2% dextrose YPAD media at 30C with shaking. Cells were collected by centrifugation and washed once with PBS and resuspended in 25mL PBS. 500μL of the cell suspension was then diluted into 50mL fresh CSM media (1.7 g/L YNB w/out ammonium sulfate, 5 g/L ammonium sulfate, 20 g/L glucose, 2 g/L SC AA mix) with 0, 1, or 10 μg/mL FLC added and allowed to grow for 5 hours at 30C with shaking. After 5 hours the OD600 was used to normalize all samples to a total cell number of ~5x10^8 cells. These cells were harvested by centrifugation, washed once in PBS, and transferred to 1.5mL Eppendorf tubes, pelleted by centrifugation, supernatant discarded, and cell pellets frozen at -80C. Cells were then shipped from Minneapolis to Stony Brook on dry ice for processing. Once there, lipid extraction and GC-MS was conducted as previously described [98]. Pellets with $5x10^8$ cells were used for lipid extraction. The dried total samples were resuspended in 100 μL chloroform added to 100 μL of BSTFA reagent (Thermo Scientific) and incubated at 70°C for 1 hour prior to GC-MS (Agilent 7890B GC–MS, Agilent 5977A MSD) analysis [88]. The retention time and mass spectral patterns of a sterol standard were used as references for lipid analysis. Sterol standards used in this study include ergosterol (Smolecule, catalog #S527372), lanosterol (Smolecule, catalog #S532452), obtusifoliol (Smolecule, catalog #S563624), zymosterol (Smolecule, catalog #S580329), 4,4-dimethyl zymosterol (Avanti catalog #700073), eburicol (Smolecule, catalog #S633611), episterol (Smolecule, catalog #S628882), and gramisterol (24-methylenelophenol or 4-methyl episterol) (Smolecule, catalog #S626191). Cholesterol (Avanti catalog # 700100) was added as an internal standard for these analyses prior to lipid extraction. The relative amount of unknown sterols was estimated based on the relative

percentage of the sterol to ergosterol peak areas in each sample. The mass spectrum of the sterol likely to be the toxic dienol was compared to the authentic standards and spectra profiles from the National Institute of Standards and Technology search database for 4-methyl episterol.

## Mouse survival study

Male CD-1 mice were inoculated via the tail vein with 2 x $10^5$ cells of *C. albicans*, using 5 mice per strain. The mice were monitored twice daily for survival by an observer who was blinded to the identity of the infecting strain. Mice that showed signs of substantial distress were humanely euthanized.

## Supporting information

**S1 Table. Strains used in this study.**
(XLSX)

**S2 Table. Plasmids and oligos used in this study.**
(XLSX)

**S3 Table. Amino acids variations in the Ksr1 protein sequence (including those at positions 189 and 272) across publicly available whole genome sequenced *C. albicans* isolates.**
(XLSX)

**S1 Fig. The sphingolipid biosynthesis pathway in *C. albicans*.** A flow chart showing the sphingolipid biosynthesis pathway in *C. albicans*, compiled from [32,34,37,87,99]. Chemical inhibitors are shown in red.
(PDF)

**S2 Fig. Properties of the evolved population.** (A) Survival curves for mice infected with the progenitor strain SC5314 (red) or with the evolved P10 population (black, AMS4058) (Mantel-Cox log-rank test p-value = 0.0018). (B) Representative microscopy images show differences in hyphal formation between the progenitor strain SC5314 (left) and the evolved population (right) after incubation at 37˚C for 2 h in rich media. (C) Read depth from whole genome sequencing normalized to average depth across the whole genome is shown for the evolved population, capped at an estimated 4 copies. Colors indicate changes in allele frequencies at heterozygous positions. Blue indicates an increase in the proportion of the "A" reference allele, while pink indicates an increase in the proportion of the "B" reference allele, signifying losses of heterozygosity. (D) Read depth from whole genome sequencing normalized to average depth across the whole genome is shown for the evolved population without a limit to the read depth for the full visualization of the CNV on Chr4. (C and D) Dots indicate positions of major repeat sequences. Blue dot indicates rDNA repeats.
(PDF)

**S3 Fig. Single colonies isolated from passages 2, 3, and 4.** (A) Read depth from whole genome sequencing normalized to average depth across the whole genome is shown for 5 single colonies selected from passage 2 of the evolution experiment. Colors indicate changes in allele frequencies at heterozygous positions. Blue indicates an increase in the proportion of the "A" reference allele, while pink indicates an increase in the proportion of the "B" reference allele, signifying losses of heterozygosity. (B) OD600 values for liquid culture growth assays are plotted over time for the wild-type progenitor (black), and the four unique genotypes recovered from passage 2. Growth in rich media without drug, with 1 μg/mL or 256 μg/mL FLC are shown. Error bars are standard errors for three replicates. Tables on the right side indicate

average $MIC_{50}$ and SMG values for these genotypes. (C) Read depth plotted as in (A) for five single colonies selected from passage 3 of the evolution experiment. (D) OD values for liquid culture growth assays are plotted over time for the wild-type progenitor (black), and the two unique genotypes recovered from passage 3. Growth in rich media alone or with 1 μg/mL FLC or 256 μg/mL FLC are shown. Error bars are standard errors for three replicates. Tables on the right side indicate average $MIC_{50}$ and SMG values for individual genotypes. (E) Read depth plotted as in (A) for the 5 single colonies selected from passage 4 of the evolution experiment. (PDF)

**S4 Fig. Overexpression of *NCP1* leads to an increase in $MIC_{50}$.** (A) Estimated copy number, calculated as read depth normalized to the rest of the nuclear genome, is plotted for 500 bp windows across the left arm of chromosome 4. The position of the *NCP1* gene is indicated in green. (B) RT-qPCR data show fold change in expression of *NCP1* relative to the wild-type background strain. Fold changes are shown for the strain containing the tet-off *NCP1* system in rich media without and with doxycycline and for the wild type strain (WT) and tet-off *NCP1* strain grown in 1 μg/mL FLC with and without doxycycline (and supplemented with iron). Black line indicates a fold change of 1 (no change) relative to the wild type strain. Error bars are propagated standard error of three replicates. (C) RT-qPCR data show fold change in expression of *NCP1* relative to the progenitor strain in the absence of FLC. Fold changes are shown for strain P3.3, which bears the Chr4 CNV in YPAD, and the wild type and P3.3 grown in 1 μg/mL FLC. (D) A heatmap showing relative growth after 24 hours for the progenitor strain and a strain containing the native *NCP1* gene under the control of a tet-off promoter system in the absence (*NCP1* on) and presence of doxycycline (*NCP1* off), as well as supplemented with iron (+Fe) to counteract synergistic effects between doxycycline and fluconazole. Yellow lines indicate MIC50 values. (PDF)

**S5 Fig. Combined effect of *NCP1* overexpression and *KSR1* LOH1.** (A) OD600 values for liquid culture growth assays are plotted over time for the wild type progenitor (black), a strain engineered to contain a tet-off-NCP1 allele (light green) and a strain engineered to contain both the tet-off-NCP1 allele and the *KSR1* LOH1 (dark green). Growth in rich media without drug, 1 μg/mL FLC, or 256 μg/mL FLC are shown. Error bars are standard error for three replicates. MIC50 and SMG values calculated at 24 and 48 hours are shown to the left (see Methods). (B) As in (A), OD values for liquid culture growth assays are plotted over time for the wild type progenitor (black), a strain engineered to contain a tet-off-*NCP1* allele (light green), and the same strain grown in the presence of doxycycline (orange). Growth in rich media, 1 μg/mL FLC, and 256 μg/mL FLC are shown. (PDF)

**S6 Fig. Heterozygous and homozygous deletion mutants of *KSR1*.** (A) A heatmap showing OD600 values relative to growth in rich media after 24 hours in a broth microdilution assay for engineered strains with homozygous or heterozygous deletions of *KSR1*. The engineered strain containing the same *KSR1* genotype that evolved in *KSR1* LOH1 is shown last for comparison. Yellow lines indicate the FLC $MIC_{50}$. (B) Read depth from whole genome sequencing normalized to average depth across the whole genome is shown for 4 single colonies with complete deletions of *KSR1*. Colors indicate changes in allele frequencies at heterozygous positions. Blue indicates an increase in the proportion of the "A" reference allele, while pink indicates an increase in the proportion of the "B" reference allele, signifying losses of heterozygosity. (C) Read depth and SNV composition from integrated genomics viewer (IGV) are shown for one evolved isolate (Evolved P10) and engineered homozygous knockout mutants in SC5314 and

in BWP17 backgrounds. All four full knockout mutants show losses of heterozygosity, indicated by full color SNVs, including *orf19.6134* and *PIF1* which are all homozygous for the reference "A" allele. *KSR1* is flanked by the uncharacterized *orf19.6123* and *MRLP8*.
(PDF)

**S7 Fig. RT-qPCR for *CDR1* in *KSR1* LOH1 strain.** (A) RT-qPCR data for one biological replicate in technical triplicate is shown for expression of the gene *CDR1* relative to the gene *ACT1* for the *KSR1* LOH1 engineered strain, the *KSR1* LOH1 engineered strain growth in 1 μg/mL FLC, and the progenitor strains grown in 1 μg/mL FLC, all relative to the WT (progenitor) strain grown in rich media. Error bars are standard deviations for technical triplicate with propagation of error during delta-delta CT calculations. (B) Data for the first biological replicate, calculated as in (A) but normalized to the gene *TEF1* rather than *ACT1*. (C) As in (A), RT-qPCR data for a second biological replicate in technical triplicate.
(PDF)

**S8 Fig. *KSR1* LOH strains show increased sensitivity to Amphotericin B.** Growth of each strain on RPMI agar plates with Amphotericin B e-test strips are shown for the wild type progenitor strain (top, AMS2401), and *KSR1* mutant strains. E-strip readings are shown to the right of the strain collection number.
(PDF)

**S9 Fig. Additional evolved strains with LOH at *KSR1*.** (A) Read depth from whole genome sequencing normalized to average depth across the whole genome for three evolved strains that all contain LOH at the *KSR1* locus and their SC5314 progenitor. Gray bars show heterozygous positions. (B) Microscopy of progenitor and evolved strains grown at 37°C for 2 hours shows differences in the initiation of hyphal growth. (C) Survival curves for *G. mellonella* show a reduction in virulence for all three evolved strains relative to the progenitor (Log-rank test, ** $P < 0.01$). (D) Median *G. mellonella* survival over 14 days demonstrates that treatment with FLC increases survival of the wild-type strain (gray) but does not increase survival for larvae infected with evolved strains LOH3 (green) and LOH4 (blue).
(PDF)

**S10 Fig. RNA-sequencing analysis for Evolved LOH2, 3, and 4 isolates.** (A) Principal component analysis for RNA-sequencing data colored according to environmental condition. (B) Principal component analysis for RNA-sequencing data colored according to strain identity. (C) A heatmap showing normalized abundance from triplicate RNA-seq data for genes annotated as being related to sphingolipid biosynthesis. Rows are genes and columns are all samples, including the wild type progenitor, Evolved LOH2, Evolved LOH3, and Evolved LOH4 in the absence of FLC (YPD), 2 μg/mL FLC, or 64 μg/mL FLC. Row and column dendrograms from k-means clustering are shown on the top and left. (D) A heatmap shows the $\log_2$ fold changes from triplicate RNA-seq data for genes catalyzing the steps of the sphingolipid biosynthesis pathway (S1 Fig). Columns are strains Evolved LOH2, Evolved LOH3, and Evolved LOH4 relative to the wild type progenitor in 0 μg/mL FLC, and rows are genes annotated as part of the sphingolipid biosynthesis pathway or related to sphingolipid transport. Fold changes that are significantly different at an adjusted p-value of 0.1 are indicated with an asterisk. (E) As in (D), a heatmap shows $\log_2$ fold changes from triplicate RNA-seq data for genes annotated as ABC transporters.
(PDF)

**S11 Fig. Whole Genome Sequencing for all engineered *KSR1* strains.** For each *KSR1* mutant engineered strain, read depth is plotted on the y-axis according to genomic position on the x-

axis. All strains shown are euploid (copy number of 2). In addition, grey bars show regions of heterozygosity, pink bars show regions of homozygosity of the "B" allele, and blue bars show homozygosity of the "A" allele. The *tetO-NCP1*, *KSR1* LOH1 strain was engineered in the SN152 genetic background, that includes LOH on Chr2 and a small region of Chr3. Points below each diagram mark the position of major repeat sequences.
(PDF)

**S12 Fig. Vacuolar phenotypes for evolved and engineered *KSR1* LOH mutants.** (A) Fluorescent microscopy (top rows) and DIC (subsequent rows) images for strains engineered to contain the *KSR1* LOH1 or to be completely homozygous for *KSR1* for either the A or B alleles. Cells were stained with the vacuolar dye FM4-64 after growth in YPD without drug (top) or after exposure to 128 μg/mL FLC (bottom). (B) Quantification of vacuolar area and integrated pixel intensity for fluorescence microscopy images shown in panel A, (see Methods). Points show an average of at least 200 cells and error bars represent standard errors of the mean (SEM). (C) As in A, fluorescence microscopy images of the additional three evolved strains with LOH affecting *KSR1*. (D) Quantification of fluorescence microscopy shown in panel (C). (B and D) Asterisks denote significant differences, using nonparametric t-tests, * P < 0.05, ** P < 0.01.
(PDF)

**S13 Fig. BODIPY staining and microscopy for evolved and engineered *KSR1* mutants.** (A) Fluorescent microscopy images for cells from strains engineered to contain the *KSR1* LOH1 and to be completely homozygous for *KSR1* for either the A or B allele. Cells were stained with the lipid droplet dye BODIPY after growth in YPD alone (top), after exposure to 128 μg/mL FLC (middle), or after exposure to 128 μg/mL myriocin (MYO, bottom). (B) Quantification of the number of lipid droplets, lipid size, and integrated pixel intensity for fluorescence microscopy images, a subset of which are shown in panel (A), (see Methods). Points are an average of at least 200 cells and error bars represent standard errors of the mean (SEM). (C) As in A, fluorescence microscopy images for the three additional evolved strains with LOH affecting *KSR1*. (D) Quantification of fluorescence microscopy, a subset of which is shown in panel (C). (B and D) Asterisks denote significant differences, using nonparametric t-tests, * P < 0.05, ** P < 0.01.
(PDF)

## Acknowledgments

We are grateful to lab members Robert T. Todd (UMN), Naomi Ziv (TAU), and Cristina Avila (TAU) for technical support. We thank Chen Bibi for constructing plasmid pJB-T510, and Dr. Anna Dukhovney for assisting with microscopy imaging.

## Author Contributions

**Conceptualization:** Pétra Vande Zande, Judith Berman, Iuliana V. Ene, Anna Selmecki.

**Data curation:** Pétra Vande Zande, Cécile Gautier.

**Formal analysis:** Pétra Vande Zande, Nora Kawar, Corinne Maufrais, Eli Isael Maciel, Caroline Mota Fernandes, Norma V. Solis.

**Funding acquisition:** Pétra Vande Zande, Maurizio Del Poeta, Scott G. Filler, Judith Berman, Iuliana V. Ene, Anna Selmecki.

**Investigation:** Pétra Vande Zande, Cécile Gautier, Nora Kawar, Corinne Maufrais, Katura Metzner, Elizabeth Wash, Annette K. Beach, Ryan Bracken, Eli Isael Maciel, Nívea Pereira de Sá, Caroline Mota Fernandes, Norma V. Solis, Maurizio Del Poeta.

**Project administration:** Anna Selmecki.

**Supervision:** Anna Selmecki.

**Validation:** Pétra Vande Zande.

**Visualization:** Pétra Vande Zande.

**Writing – original draft:** Pétra Vande Zande.

**Writing – review & editing:** Pétra Vande Zande, Scott G. Filler, Judith Berman, Iuliana V. Ene, Anna Selmecki.

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
