## [Decision Letter · Decision Letter 0]

15 Apr 2024

Dear Dr. Selmecki,

Thank you very much for submitting your manuscript "Step-wise evolution of azole resistance through copy number variation followed by KSR1 loss of heterozygosity in Candida albicans." for consideration at PLOS Pathogens. As with all papers reviewed by the journal, your manuscript was reviewed by members of the editorial board and by several independent reviewers. In light of the reviews (below this email), we would like to invite the resubmission of a significantly-revised version that takes into account the reviewers' comments.

The reviewers are generally enthusiastic about the study but there were substantial questions raised by the impact of individual genetic modifications observed in the evolved strains and their potential impact on resistance phenotypes and mechanisms. In a revision, please place particular attention to comments 2 and 3 from Rev. 1 and to comments 5 and 6 from Rev. 2.

We cannot make any decision about publication until we have seen the revised manuscript and your response to the reviewers' comments. Your revised manuscript is also likely to be sent to reviewers for further evaluation.

Sincerely,

Tobias M. Hohl

Academic Editor

PLOS Pathogens

Michal Olszewski

Section Editor

PLOS Pathogens

Michael Malim

Editor-in-Chief

PLOS Pathogens

orcid.org/0000-0002-7699-2064

The reviewers are generally enthusiastic about the study but there were substantial questions raised by the impact of individual genetic modifications observed in the evolve strains and their potential impact on resistance phenotypes and mechanisms. In a revision, please place particular attention to comments 2 and 3 from Rev. 1 and to comments 5 and 6 from Rev. 2.

Reviewer's Responses to Questions

**Part I - Summary**

Reviewer #1: The manuscript by Petra Vande Zande et al. tried to dissect the mechanisms underlying the acquired fluconazole resistance in the human fungal pathogen Candida albicans during an in vitro evolution experiment. The authors performed elegant comparative genomic analyses and discovered two sequential events, large region CNV on Chr4 and a 711bp LOH of KSR1 gene on ChrR, that both conferred the FLC tolerance and resistance. They further employed genetic approaches to explore and validate the contribution of these genomic variants on FLC resistance/tolerance. The KSR1 LOH was linked to the sphingolipid biosynthesis and the activity of ABC transporter by biochemical assays. The authors also found that the homozygous locus of a nonsense mutation, KSR1 272*/*, influenced the fluconazole susceptibility via a different mechanism. While the genomic, genetic and biochemical data support the significance of these genomic variants in FLC resistance in C. albicans, this manuscript could be further strengthened if they focus more on the main phenotypes from the evolution experiment.

Reviewer #2: This study characterizes the genomic changes that occur upon long-term exposure to subinhibitory concentrations of fluconazole with the goal of understanding how genomic plasticity contributes to antifungal resistance. This study is well designed and in general the data support the conclusions. The results provide further insight to the mechanisms through which genomic changes can drive drug resistance in C. albicans while also highlighting a novel gene associated with azole resistance, KSR1. Overall this is nice work that contributes to an important aspect of fungal disease. It would benefit from some modifications to the language used to describe drug susceptibility and some revisions to address the following comments:

Reviewer #3: The manuscript presented by Zande et al, describes a novel perspective on the evolution of fluconazole tolerance/ resistance in C. albicans. Despite the fact that various studies have explored the impact of large- and small-scale genomic variations on azole tolerance/ resistance, the perspective on the stepwise evolutionary process among the azole adapted strains, especially at the single colony level in a large “clonal” population, is lacking. Herein, authors employed whole-genomic sequencing on fluconazole adapted C. albicans isolates and by studying single colonies were able to identify novel and recurrent LOH potentially enhancing azole resistance, which increased the fluconazole MIC 512-fold when compared to the susceptible WT strain. By leveraging genetic engineering, lipidomic, RNAseq, microscopic, and phenotypic examination authors aimed at dissecting the role of LOH in KSR1 in fluconazole resistance. Additionally, authors identified that overexpression of NCP1 can also contribute to enhancing fluconazole MIC. Overall, the study is interesting.

**Part II – Major Issues: Key Experiments Required for Acceptance**

Reviewer #1: 1. Although the data related to the homozygosity of KSR1 272 */* nonsense mutation is quite interesting, this part of the data is quite distracted from the evolution experiment in this study. As the authors pointed out, none of the evolved strains with different patterns of KSR1 LOH include the homozygosity of this mutation. It is not part of the evolution events that the authors primarily aiming here. This part of data (the nonsense allele) could serve well in a separate study, such as to investigate the comprehensive function of Ksr1p in fluconazole resistance or in cell membrane integrity. The authors should reinforce the connection between this nonsense mutation and the acquired FLC resistance in the evolution experiment if they have other evidence not shown here, or could just remove this part of data from the manuscript.

2. On the other hand, the authors did not provide sufficient evidence to support the role of the homozygosity of KSR1 189R/R in FLC resistance. They only showed the fold change in MIC50 and SMG and the growth curve (Fig.3), and the sensitivity to SDS (Fig.4) of the KSR1 189R/R strain. If they claim that the LOH of this mutation is “a major driver of resistance” (line 460), they should provide more data showing the function of this mutation, such as the intracellular fluconazole concentration and the activity of the drug transporter (R6G fluorescence) of the KSR1189R/R strain, as they did with the KSR1 LOH1 strain in the Fig.2 B-D. Also, in the model cartoon presented in Fig.5C, if the authors suggested the KSR1 189R/R mutation affecting the sphingolipid composition, they need to perform lipid assays (Fig.2A) with the KSR1 189R/R strain and show the data profile resembles that of KSR1 LOH1.

3. According to Fig. S11, there is a large LOH segment of the ChrR in the KSR1 189R/R strain? If so, how would the authors conclude the increased FLC resistance of the KSR1 189R/R strain is driven by the homozygosity of the 189R/R locus? Similarly, although all carried the homozygous 189 R/R locus of KSR1, the three evolved strains LOH2/3/4 only showed modest increase of fluconazole resistance (Fig.3B). The influence of this KSR1 189R/R locus on the FLC resistance seems quite dependent on the other genomic variants. I strongly recommend the authors to construct a new point mutation strain and re-examine the phenotypes with a "cleaner" genome background, or at least state the limitation of the current strain in the result and discussion clearly.

Reviewer #2: 1. The introduction is very long; lines 91-133 could be shortened to remove excess details

2. This manuscript uses loose or incorrect terminology to describe resistance/susceptibility.

Line 82 “Resistance is measured in the clinic as…” This is an incorrect definition of clinical resistance. Resistance in the clinic is defined as the ability of an organism to grow above defined clinical breakpoints. Though resistance is also commonly used to describe a strain that is less susceptible to a compound than its parent strain, this is not the clinical definition. In addition to correcting the clinical definition, the authors should explicitly state their definition of resistance used in this work.

3. Similarly, throughout the manuscript the authors often say that strains are “more resistant” or that “CNV increase resistance by 500 fold.” Resistance itself is not a spectrum, strains are either resistant or they are not. The correct terminology to use here would be to say that strains are “less susceptible” or “500-fold less susceptible” than parent strains.

4. Using MIC assays to determine whether strains are more or less susceptible to drug treatment is appropriate, however the authors have reported differences in MIC that fall within the error of this assay. Since it is a two-fold dilution series, a two-fold difference is considered within the error of the assay and does not represent a meaningful difference. With two-fold dilutions, the limit of detection for a meaningful differences is a four-fold difference. Therefore at least a 4 fold change in MIC should be used as the threshold for resistance.

5. The conclusion that sphingolipids and azole resistance are mediated by efflux pumps is tenuous and the authors put too much stock into this as the mechanism of resistance in the KSR1 strains. Though it has been fairly well-accepted that resistance to azoles is mediated by the active efflux of azoles by CDR1 and CDR2, the data supporting this are also tenuous. The presence and absence of pumps in the membrane have a larger role in membrane homeostasis than just pumping out drugs/azoles. This is demonstrated by the observation that CDR1 or CDR2 null mutants have altered ergosterol profiles compared to wild type in the absence of fluconazole (PMID: 24435642). Moreover, these deletion strains are less susceptible to amphotericin while strains bearing overexpression of pumps are more susceptible to amphotericin, demonstrating that pumps are doing something else that is not related to pumping out fluconazole (and likely related to the sterol/lipid content or organization of the membrane) to modulate susceptibility to membrane targeting drugs. Other membrane targeting compounds such as fluphenazine also drive extremely high expression of these pumps but do not appear to be subject to efflux by the pump themselves, further supporting the role of pumps in maintaining membrane homeostasis. Even more compelling evidence is that fluphenazine and fluconazole are antagonistic in C. albicans (as one would expect based on strong induction of CDR1 by fluphenazine), however in C. glabrata where CDR1 has been shown to efflux rhodamine and thought to contribute to azole resistance, treatment of C. glabrata with fluphenazine induces CDR1 strongly, but this does not result in antagonism with fluconazole, disconnecting the expression of pumps from azole resistance under these conditions (PMID: 37702508). It is possible that pumps are actively removing fluconazole from cells and this is driving the susceptibility phenotypes. However it is probably more likely that the activity of the pumps in the evolved strain is modulating the membrane in response to changes in sphingolipids and these changes alter the susceptibility to fluconazole.

Deletion of the pumps in the KSR1 allele backgrounds would be more compelling evidence for the role of efflux in these cells, as cdr1 mutants are only moderately more susceptible to fluconazole than wild type C. albicans. If the susceptibility in these strains returns this strain to near WT levels of fluconazole susceptibility then the argument for efflux as the mechanism of resistance would be substantiated however these studies could be confounded by changes in the membrane that occur upon loss of CDR (or compensation by other pumps). Profiling the sterols and sphingolipids in all of these strains might help to build models where certain membrane states favor higher pump expression/activity. Even just profiling the sterols in addition to sphingolipids in the strains presented in this manuscript would at least determine whether sterol content differs or not, and if so, this would likely be a major player in the susceptibility to azoles.

6. Similarly, the data presented for Cy5-FLU accumulation of cells could be the result of altered uptake due to changes in membrane permeability and function rather than drug efflux. It appears the authors did perform positive controls with beauvericin, it might be helpful to show that with beauvericin treatment, the KSR1 LOH strain accumulates Cy5-FLU to the same extent as the WT after 4 hours to support efflux rather than import defects.

7. This manuscript contains a LOT of data. Most of the supporting data is essential for supporting the conclusions of the paper or strain confirmation however the ceramide data did not add much in my opinion. The authors had previously shown in Figure 1G that the progenitor with Chr 4 CNV (p4.4) does not have a fitness cost, so it is not clear why the LOH event would be suspected of masking a fitness cost due to Chr 4 CNV.

Reviewer #3: 1. Please ensure that the addition of doxycycline concentration used in the tet-off system does not have any impact on the growth of C. albicans in the absence and presence of fluconazole. Perhaps using a strong constitutive promoter, such as TEF1 promoter, would eliminate the concern of doxy impact on C. albicans growth. Along the same line, does the expression of baseline NPC1 in the absence of doxy matches that of the desired evolved strain?

2. Is it possible that the stepwise evolution is a misinterpretation of identified genotypes given that authors have explored only a limited number of single colonies in a population (5 colonies)? It could be possible that the genetic changes identified in P3 have occurred in P2, but at a lower frequency and they were not detected since authors investigated only 5 colonies (which is understandable given the limitations).

3. Why authors studied the RNAseq 24hrs post-treatment? Similarly, why lipidomic experiments were conducted 48hrs post-treatment? This could result in inconsistency, complicate the analysis, and finally preclude authors from making potential links between RNAseq and lipidomic findings. For instance, measurement of intracellular level of fluconazole and efflux using Cy5-fluconazole and R6G at 4hrs post-infection is more reasonable and consistently shows the higher expression of an ATP-dependent efflux pump, potentially Cdr1, in pumping fluconazole outside of the cell.

4. Along the previous comment, authors are recommended to individually delete CDR1, MDR1, SNQ2 in various KSR1 mutants and assess the fluconazole MIC and tolerance, intracellular fluconazole-Cy5, and R6G. This will at least elucidate which of those efflux pumps are the main driver of azole resistance/ tolerance in KSR1 variants conferring fluconazole resistance.

5. The fact that KSR1 mutants have a higher MIC could be due to the excessive level of sphingolipids, rather than impacting the proper localization of efflux pumps in the membrane. Given that efflux pumps have numerous substrates and can be activated by multitude of compounds/ metabolic intermediates impacting yeast cell survival, the elevation of the efflux pumps could be an adaptive strategy to pump those toxic intermediates outside and therefore enhance the fitness, which coincidently confers fluconazole resistance?

6. Along the same line, KSR1 LOH showed an elevated level of Sph, dhSph-1P, and PhytoSph-1P. Therefore, it would be interesting to examine the impact of addition of each intermediate individually on C. albicans growth in the absence and presence of fluconazole, fluconazole MIC and tolerance levels, and intracellular fluconazole and efflux of R6G, and filamentation.

7. CNVs have been shown to be very plastic and rapidly revert back to WT once passaged in the absence of fluconazole. Can the same thing be observed for KSR1 LOH?

8. Can authors assess the prevalence of KSR1 LOH using the publicly available C. albicans whole-genome sequence data, especially LOH on the residue of 189?

**Part III – Minor Issues: Editorial and Data Presentation Modifications**

Reviewer #1: 4. Is LOH1 the only different locus between the genomes of isolate P4.4 and P4.5? Could the difference of FLC resistance between the two isolates resulted from loci other than the Chr4 CNV and LOH1? Also, as the authors showed, the combination of NCP1 overexpression and KSR1 LOH1 in the WT strain did not recap the high MIC50 of P4.5, is it possible that genomic variants other than Chr4 CNV and LOH1, not just other genes located in the Chr4 CNV, contribute to the FLC resistance?

5. Regarding NCP1, the authors showed the change of its expression level with the tet system, with or without FLC. I wonder what is the actual expression level of NCP1 in the evolved isolates (such as P3.3 and others) with Chr4 CNV, with or without FLC? Is it comparable to that of the NCP1 overexpression strain?

6. In Fig.3B/3D, the progenitor strain (black) grew quite robustly in the presence of high concentration (256ug/ml) of FLC, which did not happen in other experiments in this study. I’m curious what happened to the progenitor strain here, and how frequent this may happen to WT SC5314?

7. In Fig. 2, all the label “KSR1 LOH” should be “KSR1 LOH1”, according to the description in the result (line 315). In Fig. 2 panel A and Fig. S8, it is hard to match the acronyms of sphingolipid species to those full names in the sphingolipid biosynthesis pathway (Fig. S1). In fact, it will help the audience to understand the data of lipid assay in Fig.2A, if the sphingolipid biosynthesis pathway is next to it.

8. In the methods session, the authors stated that the filamentation assays were done in YPD broth at 37 Celsius while shaking at 200rpm. Such conditions may promote filamentation to the level shown in Fig. S9B but are less likely to induce the strong hyphae in Fig. S2B. Please confirm the method of filamentation assay in this experiment.

9. Again in Fig. S2B, the scale bar (5mm) looks not correct. The size of C. albicans yeast cell cannot be that large.

Reviewer #2: 1. Line 137 “result in a final MIC of >256ug/mL” this should be framed in context of the progenitor strain (ex “results in a final MIC 500 fold higher than the progenitor strain”)

2. Line 390 figure is mislabeled – I think the authors are referencing 1G (p4.4 in 0ug/mL FLC) not 2B

3. Line 336 the authors refer to two outliers being removed. It should be stated in the methods what tests/criteria were used to remove these two samples, as there are other samples within the same conditions that look quite different from the others (KSR1 LOH 10ug/mL FLC; 2 samples have similar profiles and one is pretty different)

Reviewer #3: (No Response)

PLOS authors have the option to publish the peer review history of their article (what does this mean?). If published, this will include your full peer review and any attached files.

Reviewer #1: No

Reviewer #2: No

Reviewer #3: **Yes: **Amir Arastehfar
---

## [Decision Letter · Decision Letter 1]

12 Aug 2024

Dear Dr. Selmecki,

We are pleased to inform you that your manuscript 'Step-wise evolution of azole resistance through copy number variation followed by *KSR1* loss of heterozygosity in *Candida albicans*.' has been provisionally accepted for publication in PLOS Pathogens. Please re-phrase line 451 to enhance clarity of the text as requested by Rev. 2.

Best regards,

Tobias M. Hohl

Academic Editor

PLOS Pathogens

Michal Olszewski

Section Editor

PLOS Pathogens

Michael Malim

Editor-in-Chief

PLOS Pathogens

orcid.org/0000-0002-7699-2064

The authors have addressed the reviewers comments successfully. Please re-phrase line 451 as requested by Rev 2 to enhance clarity. The manuscript is now suitable for publication in PloS Pathogens.

Reviewer Comments (if any, and for reference):

Reviewer's Responses to Questions

**Part I - Summary**

Reviewer #1: The authors have addressed previous concerns, especially with the KSR1 189R/R mutant in a clean genetic background.

Reviewer #2: The reviews of this paper address the comments provided by myself and other reviewers. These edits have much improved this manuscript and the data presented within accurately support the conclusions made.

Reviewer #3: Authors have successfully and sufficinetly have addressed comments.

**Part II – Major Issues: Key Experiments Required for Acceptance**

Reviewer #1: None.

Reviewer #2: (No Response)

Reviewer #3: Not Needed.

**Part III – Minor Issues: Editorial and Data Presentation Modifications**

Reviewer #1: None.

Reviewer #2: Line 451 is confusing "strains had smaller vacuolar area than the progenitor strain but were similar in size and morphology" is this referring to cell size and morphology or vacuolar?

Reviewer #3: Not Needed.

PLOS authors have the option to publish the peer review history of their article (what does this mean?). If published, this will include your full peer review and any attached files.

Reviewer #1: No

Reviewer #2: No

Reviewer #3: **Yes: **AA

---

## [Editor Report · Acceptance letter]

22 Aug 2024

Dear Dr. Selmecki,

We are delighted to inform you that your manuscript, "Step-wise evolution of azole resistance through copy number variation followed by *KSR1* loss of heterozygosity in *Candida albicans*.," has been formally accepted for publication in PLOS Pathogens.

Best regards,

Michael Malim

Editor-in-Chief

PLOS Pathogens

orcid.org/0000-0002-7699-2064